# The Cl⁻-channel TMEM16A is involved in the generation of cochlear Ca²⁺ waves and promotes the refinement of auditory brainstem networks in mice

Alena Maul[1,2], Antje Kathrin Huebner[3], Nicola Strenzke[4], Tobias Moser[4], Rudolf Rübsamen[5], Saša Jovanovic[5†], Christian A Hübner[3*†]

[1]Neuroscience Department, Max Delbrück Center for Molecular Medicine, Berlin, Germany; [2]Faculty of Biology, Chemistry, Pharmacy, Freie Universität Berlin, Berlin, Germany; [3]Institute of Human Genetics, University Hospital Jena, Jena, Germany; [4]Institute for Auditory Neuroscience and InnerEarLab, University Medical Center Göttingen, Göttingen, Germany; [5]Institute of Biology, Faculty of Life Sciences, University of Leipzig, Leipzig, Germany

*For correspondence:
christian.huebner@med.uni-jena.de

†These authors contributed equally to this work

Competing interest: The authors declare that no competing interests exist.

**Abstract** Before hearing onset (postnatal day 12 in mice), inner hair cells (IHCs) spontaneously fire action potentials, thereby driving pre-sensory activity in the ascending auditory pathway. The rate of IHC action potential bursts is modulated by inner supporting cells (ISCs) of Kölliker's organ through the activity of the Ca²⁺-activated Cl⁻-channel TMEM16A (ANO1). Here, we show that conditional deletion of *Ano1* (*Tmem16a*) in mice disrupts Ca²⁺ waves within Kölliker's organ, reduces the burst-firing activity and the frequency selectivity of auditory brainstem neurons in the medial nucleus of the trapezoid body (MNTB), and also impairs the functional refinement of MNTB projections to the lateral superior olive. These results reveal the importance of the activity of Kölliker's organ for the refinement of central auditory connectivity. In addition, our study suggests the involvement of TMEM16A in the propagation of Ca²⁺ waves, which may also apply to other tissues expressing TMEM16A.

## Editor's evaluation

This study addresses the extremely interesting question of how spontaneous activity in the cochlea prior to hearing onset impacts the development of auditory circuits in the brainstem. The study has many strengths, including the use of complementary in vitro and in vivo recording techniques to characterize both peripheral and central defects resulting from conditional deletion of the gene for the chloride channel TMEM16A. The reviewers identified some concerns over the interpretation of the data, but all of these concerns were addressed in the subsequent revisions.

## Introduction

Before hearing onset around postnatal day 12 (P12) in mice (*Müller et al., 2019*; *Shnerson and Pujol, 1981*; *Sonntag et al., 2009*), the afferent auditory system exhibits cochlea-driven spontaneous activity (*Jones et al., 2007*; *Lippe, 1994*). Inner hair cells (IHCs) fire bursts of action potentials (*Kros et al., 1998*), which drive afferent transmission to spiral ganglion neurons (SGNs) (*Beutner and Moser, 2001*; *Glowatzki and Fuchs, 2002*) and thus trigger bursting discharges through ascending auditory pathways (*Babola et al., 2018*; *Tritsch and Bergles, 2010*; *Tritsch et al., 2007*). Similar

to developing motor and visual systems (*Hanson and Landmesser, 2004*; *Katz and Shatz, 1996*), patterned activity of auditory neurons was proposed to promote activity-dependent refinement of auditory circuits before hearing onset (*Clause et al., 2014*; *Clause et al., 2017*).

In the developing inner ear, non-sensory inner supporting cells (ISCs) form a transient epithelial structure known as Kölliker's organ (*Hinojosa, 1977*; *Hou et al., 2019*). ATP released from ISCs through connexin hemichannels (*Mazzarda et al., 2020*) activates purinergic receptors in a paracellular manner, leading to cell volume decrease of ISCs and cochlear Ca$^{2+}$ transients (*Babola et al., 2018*; *Tritsch and Bergles, 2010*; *Tritsch et al., 2007*). It was proposed that the Ca$^{2+}$-activated Cl$^-$-channel TMEM16A, which is expressed in ISCs, might be the pacemaker for spontaneous cochlear activity (*Yi et al., 2013*). Indeed, spontaneous osmotic cell shrinkage was shown to be mediated by TMEM16A-dependent Cl$^-$ efflux, which forces K$^+$ efflux from ISCs and thus the transient depolarization of IHCs (*Yi et al., 2013*). Thereby, bursting activity of nearby IHCs, which will later respond to similar sound frequencies, becomes synchronized (*Eckrich et al., 2018*; *Harrus et al., 2018*; *Wang et al., 2015*), establishing a possible scenario for tonotopic map refinement in central auditory structures.

Using *Ano1* conditional knockout mice, we show that TMEM16A not only modulates ISC volume but also drives the amplification of localized Ca$^{2+}$ transients to propagating Ca$^{2+}$ waves within the cochlea. Prior to hearing onset, knockout mice show reduced burst firing of neurons in the medial nucleus of the trapezoid body (MNTB), downstream of SGNs and neurons of the cochlear nucleus. Moreover, the frequency selectivity of individual MNTB neurons is diminished shortly after hearing onset (P14) pointing toward reduced refinement of auditory connections. Indeed, neurons from the lateral superior olive (LSO) received twice as many functional MNTB afferents in knockout mice compared to wildtype littermates. Taken together, these results suggest that the Ca$^{2+}$-activated Cl$^-$-channel TMEM16A plays a significant role in the propagation of Ca$^{2+}$ waves and contributes to the refinement of auditory brainstem circuitries prior to hearing onset.

## Results

### TMEM16A is required for the generation of cochlear Ca$^{2+}$ waves

The Ca$^{2+}$ activated Cl$^-$-channel TMEM16A is expressed in ISCs of Kölliker's organ (for a schematic representation of a part of the organ of Corti, see *Figure 1A*; for a differential interference contrast [DIC] image, see *Figure 1B*; *Wang et al., 2015*; *Yi et al., 2013*) and is activated by an ATP-induced increase in Ca$^{2+}$ concentration (*Wang et al., 2015*; *Yi et al., 2013*). The opening of TMEM16A triggers Cl$^-$ efflux, followed by K$^+$ efflux and cell shrinkage. The ensuing rise of extracellular K$^+$ drives electrical activity of immature IHCs (*Wang et al., 2015*). To assess the role of TMEM16A in the developing cochlea and the impact of TMEM16A-dependent cochlear signaling on the development of auditory brainstem nuclei, we disrupted *Ano1* in the inner ear. This was achieved by mating our floxed *Ano1* line (*Ano1$^{fl/fl}$*) (*Heinze et al., 2014*) with a line expressing Cre-recombinase under the control of the *Pax2* promoter (*Ohyama and Groves, 2004*), which is active in the otic placode (*Lawoko-Kerali et al., 2002*). This line is subsequently referred to as cKO mice. *Ano1* deletion was confirmed by immunohistochemistry (*Figure 1—figure supplement 1A*) and Western blot analysis (*Figure 1—figure supplement 1B*). Importantly, organs of Corti of cKO mice showed no obvious morphological defects. The development of the tectorial membrane and the morphology of the hair cells appeared normal before hearing onset (P6), at hearing onset (P12), or in the weeks thereafter (3 weeks and 6 weeks after birth) (*Figure 1—figure supplement 1C and D*), indicating that TMEM16A and TMEM16A-dependent activity of Kölliker's organ is not essential for the morphological development of the organ of Corti.

To investigate the volume changes and Ca$^{2+}$ waves that spontaneously appear in the ISCs of Kölliker's organ (*Anselmi et al., 2008*; *Tritsch and Bergles, 2010*; *Tritsch et al., 2007*), acutely isolated cochleae from P5–7 wildtype and cKO littermates were used. While wildtype cochleae showed notable volume changes of groups of ISCs (*Figure 1C, G and H*; n = 7; mean event area ± SEM = 7021 ± 1128 µm$^2$; mean event frequency ± SEM = 0.0171 ± 0.0056 Hz), volume changes were almost absent in cochleae acutely isolated from 5- to 7-day-old cKO mice (*Figure 1D, G and H*; n = 9; mean event area ± SEM = 40 ± 30 µm$^2$, p=0.0000056; mean event frequency ± SEM = 0.0003 ± 0.0002 Hz, p=0.0042), in agreement with previously published results (Wang et al., 2015). In wildtype cochleae, these events propagated in waves along the tonotopic axis of the cochlea also affecting phalangeal cells that surround IHCs (*Video 1*). To visualize changes in intracellular Ca$^{2+}$ concentrations, cochlear

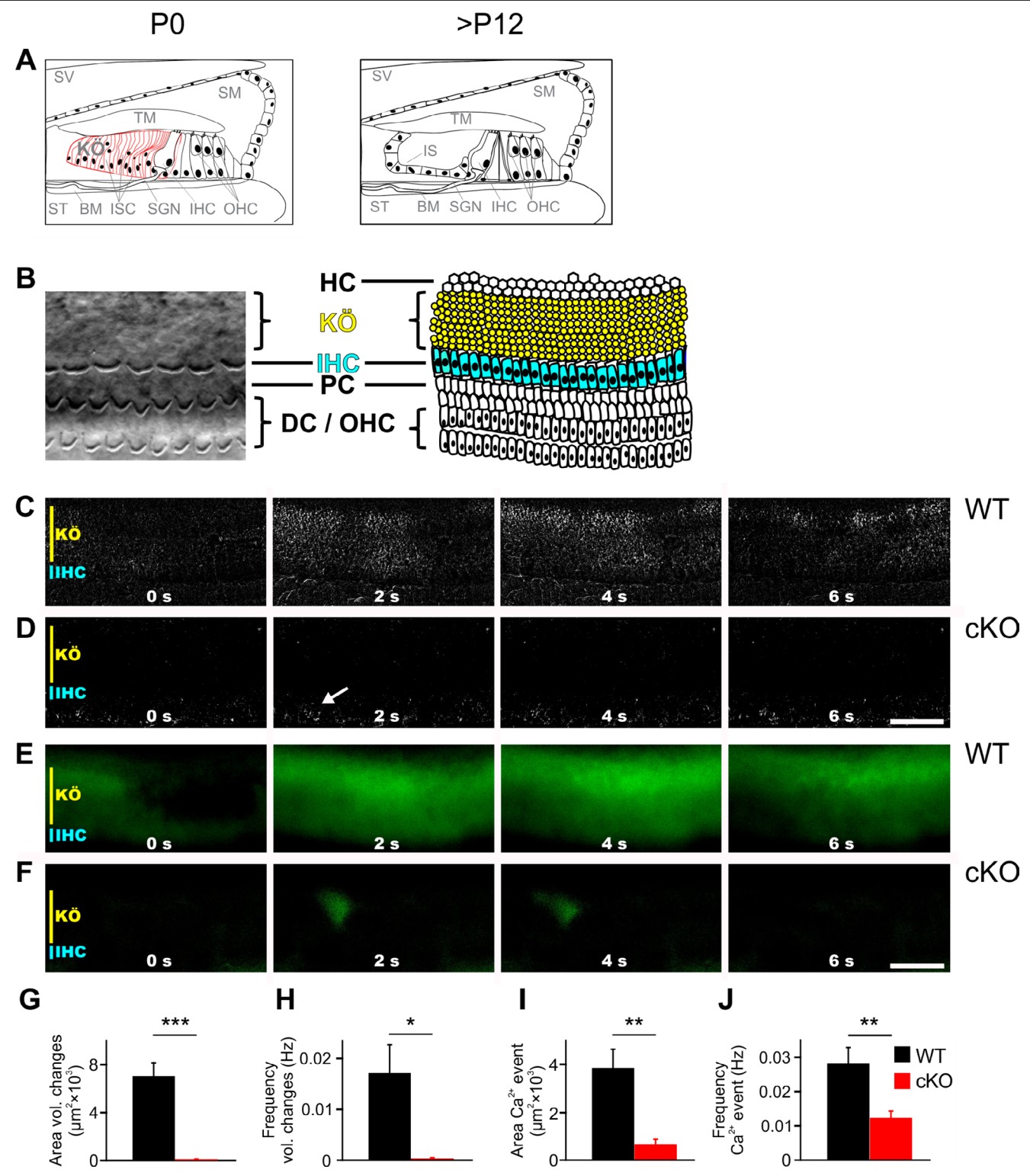

**Figure 1.** TMEM16A is required for the generation of spontaneous volume changes and Ca²⁺ waves in Kölliker's organ. (**A**) Schematic representation of the organ of Corti at birth (left) and after hearing onset (right). BM, basilar membrane; IHC, inner hair cells; IS, inner sulcus; ISC, inner supporting cells; KÖ, Kölliker's organ; OHC, outer hair cells; SGN, spiral ganglion neurons; SM, scala media; ST, scala tympani; SV, scala vestibuli; TM, tectorial membrane. (**B**) Left: example differential interference contrast (DIC) image from the area of the cochlea turn imaged in (**C–F**). Right: the schematic drawing highlights the location of IHCs (blue) and ISCs (yellow) of KÖ. HC, Hensen cells; DC, Deiter's cells; PC, pillar cells. (**C, D**) DIC time-lapse imaging at P7 reveals spontaneous volume changes of ISCs in a wildtype mouse indicated by changes in light intensity, which are almost absent in the

*Figure 1 continued on next page*

*Figure 1 continued*

cKO littermate. The arrow in (**D**) indicates erythrocytes moving in a blood vessel. Scale bar 50 μm. (**E, F**) Ca²⁺ imaging at P6 reveals spontaneous Ca²⁺ waves traveling across ISCs of KÖ in a wildtype mouse (**E**) that are reduced to small local Ca²⁺ transients in the cKO littermate (**F**). Scale bar 50 μm. (**G, H**) Quantification of areas and frequencies of spontaneous ISC volume changes. A time-lapse series of 1200 images with one image per second was analyzed. Values represent mean ± SEM (P5–7; n = 7 WT, n = 9 cKO; two-tailed unpaired Student's *t*-test: area: p=0.0000056, frequency: p=0.0042). (**I, J**) Quantification of area and frequency of spontaneous Ca²⁺ events. A time-lapse series of 400 images with one image per second was analyzed. Values represent mean ± SEM (P5–7; n = 14 WT, n = 16 cKO; two-tailed unpaired Student's *t*-test: area: p=0.00032; frequency: p=0.0027).

The online version of this article includes the following source data and figure supplement(s) for figure 1:

**Source data 1.** Source data for *Figure 1*.

**Figure supplement 1.** Comparison of morphology and TMEM16A expression patterns in the developing organ of Corti between wildtype and cKO mice.

**Figure supplement 2.** Absence of P2 receptor activation in cKO mice.

**Figure supplement 2—source data 1.** Source data for *Figure 1—figure supplement 2*.

explants were loaded with the Ca²⁺ indicator dye Fura-2 AM. Wildtype mice showed Ca²⁺ transients that traveled in waves along the tonotopic axis of Kölliker's organ (*Figure 1E, I and J*; n = 14; mean event frequency ± SEM = 0.0282 ± 0.0050 Hz; mean event area ± SEM = 3839 ± 794 μm²). In cochleae from cKO littermates, Ca²⁺ transients were rare and restricted to small areas (*Figure 1F, I and J*, *Video 2*; n = 16; mean event frequency ± SEM = 0.0123 ± 0.0020 Hz; p=0.0027; mean event area ± SEM = 656 ± 214 μm²; p=0.00032).

Spontaneous Ca²⁺ waves are elicited by ATP-induced ATP release from ISCs (*Mazzarda et al., 2020*), a mechanism possibly involving P2Y1,

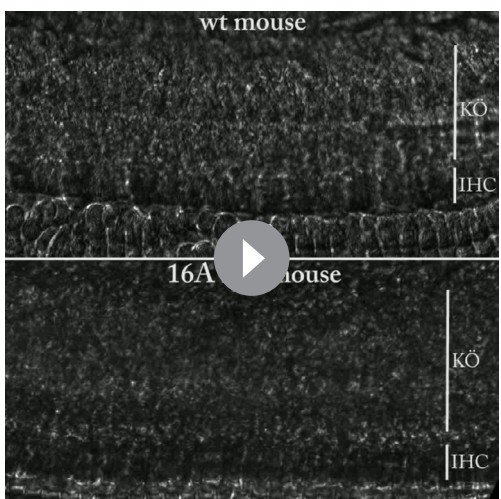

**Video 1.** TMEM16 A is required for the generation of spontaneous volume changes in Kölliker's organ (KÖ). Time-lapse imaging (one image per second) reveals spontaneous volume changes of inner supporting cells of in a wildtype mouse cochlea (P7), which propagate in a wave-like manner up and down the cochlea turn. In contrast, volume changes are almost absent in the cochlea isolated from a cKO littermate (the bottom of the video shows erythrocytes moving in a blood vessel). Images were processed using a custom-written ImageJ macro and the ImageJ software. Each frame was subtracted from an average of five preceding frames to highlight the changes in light scattering caused by the changes in cell volume. The video (seven images per second) shows the top view of an area from an isolated cochlea turn. KÖ, Kölliker's organ; IHC, inner hair cells.
https://elifesciences.org/articles/72251/figures#video1

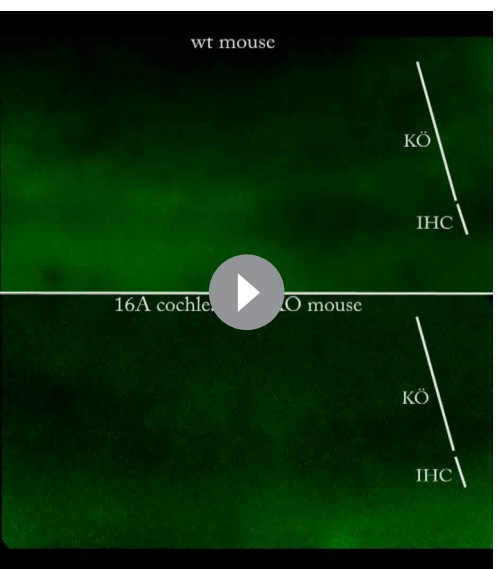

**Video 2.** TMEM16A is required for the propagation of cochlear Ca²⁺ waves. Time-lapse imaging (one image per second) reveals spontaneous Ca²⁺ signals in the inner supporting cells that propagate up and down the cochlear turn in a wildtype mouse cochlea (P7). In contrast, Ca²⁺ waves are reduced to local Ca²⁺ events in the cochlea of a cKO littermate. Images were processed using a custom-written ImageJ macro and the ImageJ software. Each frame was subtracted from an average of five preceding frames to highlight the changes in Ca²⁺ concentration. The video (seven images per second) shows the top view of an area from an isolated cochlea turn. KÖ, Kölliker's organ; IHC, inner hair cells.
https://elifesciences.org/articles/72251/figures#video2

P2Y2, and P2Y4 receptors (*Babola et al., 2018*; *Piazza et al., 2007*), Cx26 and Cx30 heteromeric hemichannels (*Anselmi et al., 2008*; *Mazzarda et al., 2020*; *Schütz et al., 2010*), and TMEM16A (*Wang et al., 2015*). To assess what role TMEM16A might play in ATP-mediated $Ca^{2+}$ signals, we applied the nonselective P2 receptor antagonist suramin (150 µM) (*Burnstock, 2014*; *von Kügelgen and Wetter, 2000*). In wildtype mice, suramin application reduced $Ca^{2+}$ waves to uncoordinated, locally restricted $Ca^{2+}$ transients (n = 6; mean event area before suramin application ± SEM = 4476 ± 1007 µm² and after suramin application ± SEM = 1172 ± 389 µm²; p=0.0324 paired two-tailed Student's *t*-test). Suramin application had no effect on $Ca^{2+}$ transients in cKO mice (n = 5; mean event area before suramin application ± SEM = 466 ± 172 µm² and after suramin application ± SEM = 379 ± 103 µm²; p=0.4842 paired two-tailed Student's *t*-test) (*Figure 1—figure supplement 2*). This supports the notion that TMEM16A is important for the propagation of spontaneous activity between ISCs of Kölliker's organ, probably via P2 receptors.

## TMEM16A-dependent cochlear activity modulates the burst-firing pattern of MNTB neurons

By increasing the $K^+$ concentration, TMEM16A-dependent activity of Kölliker's organ leads to the generation of $Ca^{2+}$ action potentials in IHCs. This is followed by $Ca^{2+}$-dependent exocytosis of glutamate at the IHC synapse, which drives burst firing of action potentials in SGNs (*Wang et al., 2015*). The bursting activity is then relayed to central auditory neurons (*Babola et al., 2018*; *Tritsch and Bergles, 2010*; *Figure 2A* shows a schematic representation of the auditory brainstem) and is believed to be important for the proper development of synaptic contacts and tonotopic maps (*Clause et al., 2014*; *Clause et al., 2017*).

To investigate the effects of *Ano1* knockout on the burst-firing patterns in auditory brainstem neurons, juxtacellular single-unit recordings from MNTB neurons were obtained from in vivo prehearing mice (P8). Similar to SGNs (*Tritsch et al., 2010*; *Jing et al., 2013*), MNTB neurons exhibit spontaneous bursts of spikes that could last several seconds (*Figure 2—figure supplement 1A*; *Sonntag et al., 2009*). These long bursts are made up of a series of short 'mini-bursts consisting of several spikes and occurring at intervals of approximately 100 ms (*Figure 2—figure supplement 1B and C*).

Bursting activity was quantified by the coefficient of variation (CV) of interspike intervals (ISIs), whereby values below 1 correspond to random firing and higher values indicate more patterned activity (*Jones et al., 2007*). In wildtype mice, MNTB neurons showed the typical bursting activity (*Figure 2B and F*; n = 14, median CV [25%, 75% quartiles] = 3.2 [2.65, 3.56]) (*Sonntag et al., 2009*), while in cKO littermates MNTB firing patterns had significantly smaller CVs (*Figure 2C–F*; n = 15, median CV [25%, 75% quartiles] = 1.7 [1.2, 2.9]; p=0.006, Mann–Whitney test). Intermediate ISIs, representing the interval between mini-bursts, were severely diminished in all MNTB neurons recorded from cKO mice. A presentation of the firing pattern as mean cumulative distribution of ISIs (*Figure 2G*; Kolmogorov–Smirnov test: p=0.0008, D = 0.19) and in an overlaid log-binned histogram (*Figure 2H*; chi-square test, p-values are shown in *Supplementary file 1a*) illustrates a shift toward longer ISIs in cKO mice. The additional detailed analysis of burst-firing patterns in MNTB neurons revealed further significant differences between wildtype and cKO littermates for (i) the number of bursts per 100 s (median [25%, 75% quartiles]: WT = 3.9 [3.3, 5.8]; cKO = 0.6 [0, 3.8]; p=0.009, Mann–Whitney test), (ii) the number of spikes per burst (median [25%, 75% quartiles]: WT = 45 [21, 89]; cKO = 19 [14, 31]; p=0.00001, Mann–Whitney test), and (iii) the duration of bursts (median [25%, 75% quartiles]: WT = 1.6 [0.9, 3.3]; cKO = 0.7 [0.3, 1.8]; p=0.000002, Mann–Whitney test) (*Figure 3A–C*). MNTB neurons recorded from cKO mice showed a large variation of action potential firing rates, but still the mean firing rate did not differ from wildtype cells (*Figure 3D*; mean firing rate ± SEM: WT = 6.2 ± 0.8 Hz, n = 14; cKO = 5.8 ± 1.1 Hz, n = 15; p=0.73, Student's *t*-test). Also, firing rates within bursts did not differ between TMEM16A cKO and WT mice (data not shown; WT: 26 [19; 36.8] AP/s, n = 85; cKO: 32 [16.4; 48.8] AP/s, n = 60; p=0.34, Mann–Whitney rank-sum test).

Since TMEM16A is neither expressed in SGNs nor in CN, MNTB, and LSO neurons in wildtype mice and expression in the brainstem was limited to vascular smooth muscle cells (*Figure 3—figure supplement 1*), we primarily attribute the severely altered burst-firing activity to impaired Kölliker's organ activity in cKO mice.

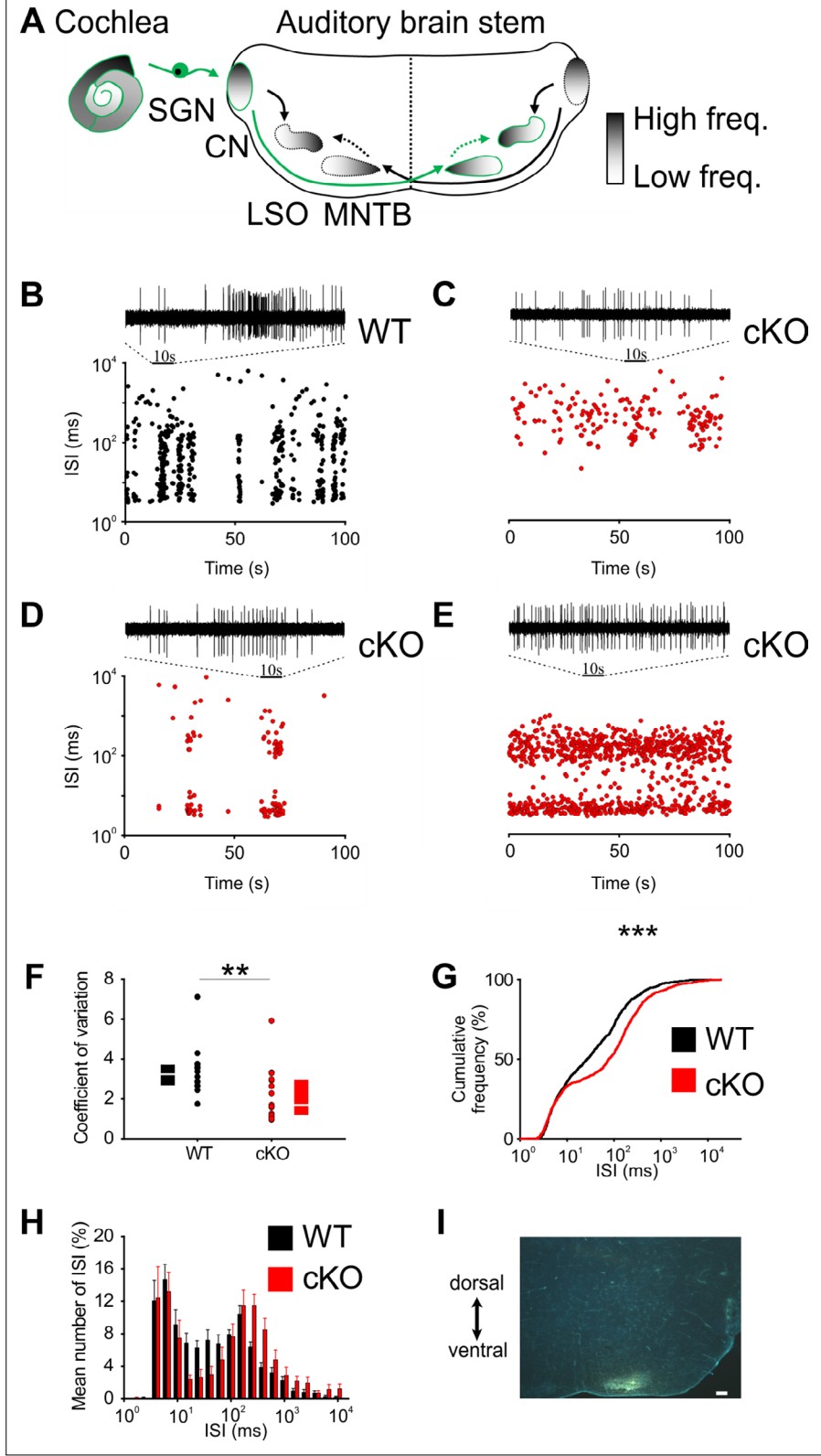

**Figure 2.** Disruption of Kölliker's organ activity changes prehearing burst firing of medial nucleus of the trapezoid body (MNTB) neurons in vivo. (**A**) Simplified model of auditory connections in the brainstem. The pathways relevant for this experiment are marked in green. Inhibitory pathways are indicated by dotted arrows. CN, cochlear nucleus; LSO, lateral superior olive; SGN, spiral ganglion neurons. (**B–E**) Patterns of spontaneous discharge activity

*Figure 2 continued on next page*

*Figure 2 continued*

from individual MNTB neurons recorded from mice before hearing onset (P8) in wildtype and cKO littermates. Dotplot graphs show respective interspike interval (ISI) distributions for 100 s of spontaneous discharge activity. On top of each dotplot raster is a 10 s period of original spike trains. Note that the wildtype MNTB neuron shows prominent burst firing, which is either strongly reduced (**D**) or absent in cKO mice (**C, E**). (**F**) Quantification of spike bursting patterns by calculating the coefficient of variation of ISIs yields significant differences between wildtype (n = 14) and cKO units (n = 15) (Mann–Whitney rank-sum test: p=0.006); also shown are boxplots indicating medians and 25% and 75% quartiles. (**G**) The mean cumulative distribution of ISIs reveals the significant shift toward larger values in cKO mice (wildtype n = 14 and cKO n = 15; Kolmogorov–Smirnov test: p=0.0008, D = 0.19), with the median ISI increasing from 26.9 ms in wildtype to 76.3 ms in cKO mice. (**H**) The overlaid log-binned histogram compares the distribution of ISIs between wildtype and cKO mice. Values represent mean ± SEM (n = 14 wildtype, n = 15 cKO [P8]). For statistical analysis, the chi-square test was used (see *Supplementary file 1a* for p-values). (**I**) Iontophoretic injection with Fluorogold verifies recording site from in vivo juxtacellular voltage recordings from MNTB neurons in a prehearing wildtype mouse (P8). Scale bar 200 µm.

The online version of this article includes the following source data and figure supplement(s) for figure 2:

**Source data 1.** Source data for *Figure 2A–G*.

**Source data 2.** Source data for *Figure 2H*.

**Figure supplement 1.** Spontaneous activity of an exemplary P8 medial nucleus of the trapezoid body (MNTB) wildtype neuron in vivo.

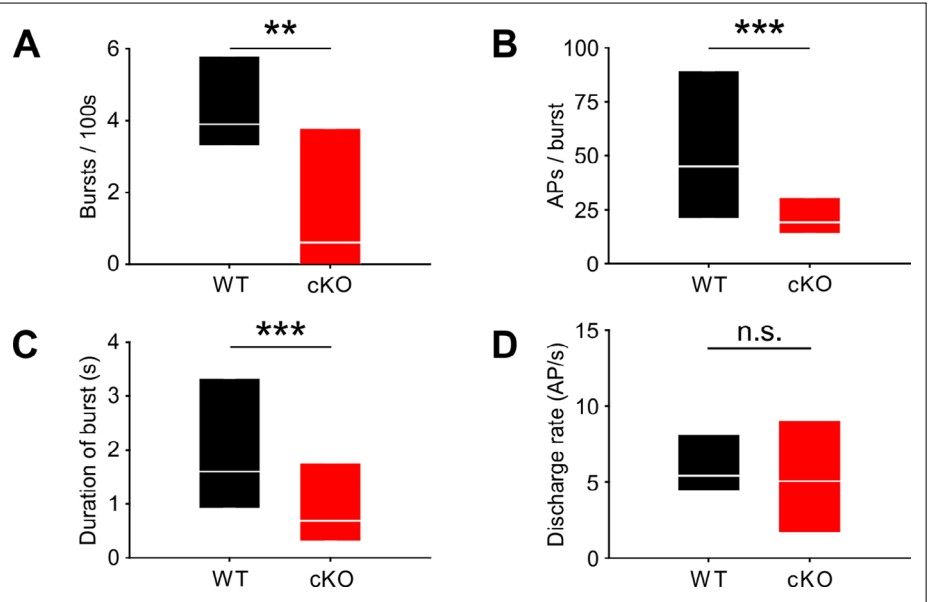

**Figure 3.** Lack of TMEM16A in cochlear inner supporting cells (ISCs) changes bursts but not the overall firing rate in medial nucleus of the trapezoid body (MNTB) auditory brainstem neurons in vivo. (**A–C**) Spontaneous discharge patterns of MNTB neurons recorded from cKO mice show a reduced number of bursts per 100 s (**A**), a reduced number of spikes per burst (**B**), and a reduced duration of bursts (**C**) compared to wildtype. Values represent median with 25%, 75% quartiles (n = 14 WT; n = 15 cKO [P8]; Mann–Whitney rank-sum test: number of bursts per 100 s: p=0.009; number of spikes per burst: p=0.00001; duration of burst: p=0.000002). (**D**) The overall discharge rates did not differ between wildtype (n = 14) and cKO (n = 15) (two-tailed unpaired Student's *t*-test: p=0.73).

The online version of this article includes the following source data and figure supplement(s) for figure 3:

**Source data 1.** Source data for *Figure 3*.

**Figure supplement 1.** TMEM16A is not expressed in spiral ganglion (SG), cochlear nucleus (CN), medial nucleus of the trapezoid body (MNTB), and lateral superior olive (LSO) neurons before hearing onset in wildtype mice.

## Frequency selectivity of MNTB neurons is reduced in cKO mice

To test whether the changes in burst-firing patterns of cKO MNTB neurons have consequences on neuronal function after hearing onset, auditory brainstem responses (ABRs) were measured at P13–14. cKO mice had similar ABR thresholds in response to stimulation with clicks or tone bursts at 6, 12, and 24 kHz as wildtypes (*Figure 4A and B*, *Supplementary file 1b*; n = 6 WT; n = 7 cKO). In both genotypes, ABRs to click stimuli of various intensities (40–100 dB) mainly consisted of three waves (labeled I–III), which were comparable in latency and amplitude (*Figure 4—figure supplement 1*, *Supplementary file 1c and d*). These data indicate that cKO mice have a normal sensitivity to sound stimulation and normal temporal precision of the spiking response to sound onset in the lower auditory pathway.

Next, we assessed whether the disruption of TMEM16A-dependent cochlear activity affects the frequency tuning properties of MNTB neurons. Therefore, the frequency response areas (FRAs) of single MNTB neurons were acquired in four cKO and four wildtype littermates using in vivo electrophysiology and tone burst stimulation. Juxtacellular recordings were performed at P14, that is, shortly after the onset of hearing to avoid possible compensatory effects of acoustically driven activity on neuron responsiveness (*Werthat et al., 2008*; *Bogart et al., 2011*). The characteristic frequencies (CFs) of the recorded MNTB neurons, that is, the frequency value at which the neuron is excited with the lowest intensity , ranged between 5.3 and 30.5 kHz and did not differ between the two groups (mean CF ± SEM: WT = 15.4 ± 1.1 kHz [n = 25]; cKO = 16.2 ± 1.3 kHz [n = 32]; p=0.51, Student's *t*-test). MNTB neurons recorded from wildtype mice showed the typical V-shaped FRAs with acoustically driven excitation sharply narrowing toward lower intensities (*Figure 4C*). The filter characteristics of the FRAs were quantified by the $Q_n$-value, a measure of the unit's sharpness of tuning, which is calculated as the ratio of CF to bandwidth at 10, 20, and 30 dB above threshold (e.g., $Q_{10}$ = CF/$BW_{10}$ with $BW_{10}$ = bandwidth at 10 dB above threshold). For wildtype mice, the median [25%, 75% quartiles] was $Q_{10}$ = 5.5 [4.7, 9.2], $Q_{20}$ = 4.6 [3.8, 6.8], and $Q_{30}$ = 3.6 [3.4, 5] (*Figure 4E*). Neurons recorded from the cKO littermates (n = 32) had significantly broader excitatory response areas, that is, lower frequency selectivity as indicated by significantly lower Q-factors at all three above-threshold levels (*Figure 4D and E*; median [25%, 75% quartiles]: $Q_{10}$ = 4.7 [2.8, 6.3], p=0.03; $Q_{20}$ = 3.1 [2.5, 4.9], p=0.008; $Q_{30}$ = 2.7 [2.0, 3.7], p=0.002, Mann–Whitney test). Additionally, MNTB neurons in cKO mice had elevated thresholds in comparison to the wildtype littermates (median threshold [25%, 75% quartiles]: cKO: 6.3 [0.8, 29.6] dB SPL; WT = 0.6 [0.0, 4.6] dB SPL; p=0.006, Mann–Whitney test) (*Figure 4F*). Overall CF threshold levels tended to show a larger variability in knockout compared to wildtype mice (cKO: –7.3 dB SPL to 48.3 dB SPL; WT: –8.4 dB SPL to 18.7 dB SPL). Furthermore, the sound-evoked firing properties of the MNTB neurons in cKO mice were also affected. The rate-level functions at CF showed significantly lower firing rates for sound intensities at and above 10 dB SPL of tone-burst stimulation in comparison to wildtype littermates (effect of strain p<0.001, effect of intensity p<0.001, interaction strain × intensity p=0.006; two-way ANOVA) (*Figure 4G*). The maximal firing rate of individual neurons in response to any CF/intensity combination was markedly diminished in cKO mice (mean FR ± SEM: WT = 263.6 ± 12.4 action potentials/s, n = 25; cKO = 223.9 ± 11.1 action potentials/s, n = 32; p=0.015, *t*-test) (*Figure 4H*). Apparently, MNTB neurons in cKO mice achieve rates that are typically observed in wildtype littermates. Taken together, these data demonstrate that frequency selectivity and sensitivity to acoustic stimulation in single MNTB neurons are impaired upon disruption of *Ano1* in the cochlea.

## The developmental refinement of functional connections of the MNTB-LSO pathway is impaired in cKO mice

Despite the above-described differences in the pattern of spontaneous and sound-evoked activity in MNTB neurons between wildtype and cKO mice, the gross morphology of the MNTB and the LSO appeared normal (mean MNTB area ± SEM: WT = 63 × $10^3$ ± 4.6 μm² [n = 10]; KO = 55 × $10^3$ ± 3.4 μm² [n = 10]; p=0.0831; mean LSO area ± SEM: WT = 117 × $10^3$ ± 6.8 μm² [n = 10]; KO = 110 × $10^3$ ± 3.9 μm² [n = 10]; p=0.4325).

Since spontaneous burst activity of auditory neurons might promote the targeting and refinement of their projections as suggested for the developing visual system (*Torborg et al., 2005*), we assessed the synaptic and topographic refinement of MNTB and LSO neurons. MNTB neurons were activated via photolysis of caged glutamate. A 405 nm continuous diode laser was used for illumination. Laser flashes were delivered through a light guide of 20 μm diameter, which produced circular spots of 20 μm diameter in the focal plane. Laser pulses of 10 ms duration were delivered with 6 s delay time

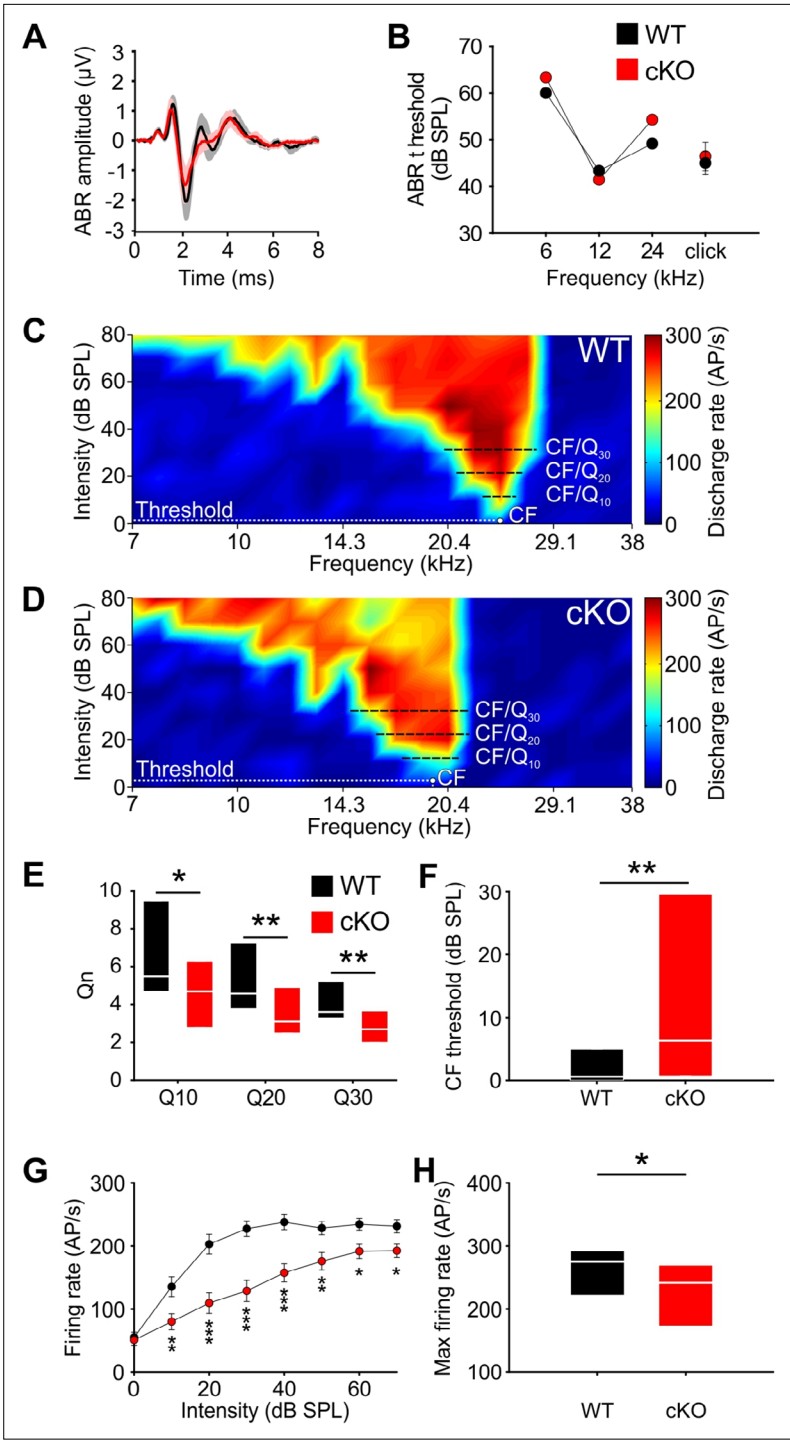

**Figure 4.** Wildtype and cKO mice show similar auditory brainstem response (ABR) thresholds, but differences in frequency selectivity, response threshold, and maximal firing rate in neurons of the medial nucleus of the trapezoid body (MNTB). (**A**) Grand averages of ABR waveforms to 80 dB click stimulation recorded from cKO (red) and wildtype (black) at P13–14. (**B**) ABR thresholds in response to 6, 12, or 24 kHz tone bursts and click stimuli did not differ between wildtype (n = 7) and cKO (n = 6) at P13–14. For values, see ***Supplementary file 1b***. (**C, D**) Representative frequency response areas of MNTB neurons (P14) recorded juxtacellularly in a wildtype mouse (**C**) (characteristic frequencies [CF]: 24 kHz, threshold 0.1 dB SPL, Q10/20/30 = 6.4/4.3/4.2), and in a cKO littermate (**D**) (CF: 18.4 kHz, threshold = 3 dB SPL, Q10/20/30 = 3.7/3.2/2.8). Note that the response area is broader and that the CF thresholds are increased in cKO. (**E**) Frequency selectivity of MNTB neurons was reduced in cKO mice as indicated by lower Q-factors (shown are medians and 25%, 75% quartiles [n = 25 WT; n = 32 cKO, P14]; Mann–

*Figure 4 continued on next page*

*Figure 4 continued*

Whitney rank-sum test: Q10: p=0.03, Q20: p=0.008, Q30: p=0.002). (**F**) Sound thresholds of individual MNTB neurons are elevated in cKO mice compared to wildtype (p=0.006). (**G**) Average rate-level functions at CF show decreased action potential firing in cKO at SPLs above 10 dB (two-way ANOVA: effect of strain p<0.001, effect of intensity p<0.001). (**H**) Maximum firing rates during acoustic stimulation are significantly decreased in cKO mice compared to wildtype littermates (two-tailed unpaired Student's *t*-test: p=0.015).

The online version of this article includes the following source data and figure supplement(s) for figure 4:

**Source data 1.** Source data for *Figure 4A and B*.

**Source data 2.** Source data for *Figure 4E–H*.

**Figure supplement 1.** Wildtype and cKO mice have similar auditory brainstem responses (ABRs).

between uncaging sites. The number of functional inputs on individual LSO neurons was assessed by whole-cell current-clamp recordings. In a pilot experiment, the distance was measured at which glutamate uncaging produces action potentials in MNTB neurons. The mean distance (± SEM) was 19.2 ± 0.8 μm mediolaterally and 20.0 ± 1.3 μm dorsoventrally, indicating that glutamate uncaging was locally restricted to a surface area of 20 × 20 μm² and that light scattering that might influence the amount of uncaged glutamate in the tissue was negligible (*Figure 5—figure supplement 1A–C*).

Notably, the size of MNTB input areas in cKO mice was doubled compared to wildtype (*Figure 5A–C*; mean cross-sectional input area ± SEM: WT = 10% ± 1% [n = 10] and cKO: 20% ± 3% [n = 10] of the respective MNTB cross-sectional areas; p=0.0017, Student's *t*-test). Moreover, the input width,

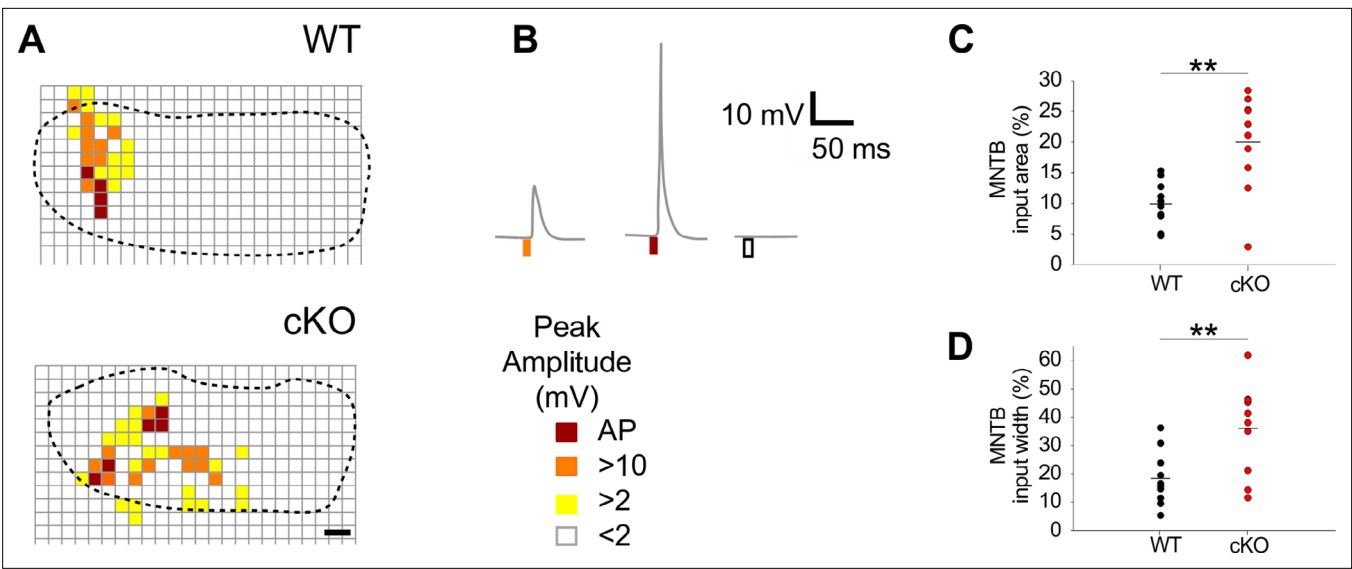

**Figure 5.** Medial nucleus of the trapezoid body-lateral superior olive (MNTB-LSO) input maps are enlarged upon disruption of TMEM16A.
(**A**) Exemplary MNTB input maps from wildtype (P9) and cKO mice (P10) as revealed by whole-cell current-clamp recordings (dotted line outlines the MNTB area, grid points indicate glutamate uncaging sites). The location of responsive (colored squares) and unresponsive (open squares) uncaging sites is indicated. Scale bar 40 μm. (**B**) Uncaging of glutamate close to presynaptic MNTB neurons gives rise to synaptic responses of various peak amplitudes (left), elicits action potentials (middle), or fails to evoke significant voltage signals (right) in the recorded LSO neurons (see color code below). (**C, D**) Quantification of MNTB input areas (summed area of all responsive uncaging sites) on a recorded LSO neuron and of the MNTB input widths (maximal distance of stimulations sites that evoked depolarization greater than 10 mV along the mediolateral [tonotopic] axis of the MNTB). MNTB input areas were normalized to the corresponding MNTB cross-sectional area and MNTB input widths to the length of the mediolateral axis of the MNTB. Values represent mean ± SEM (n = 10 WT; n = 10 cKO [P9–11]; two-way unpaired Student's *t*-test: MNTB input area: p=0.0017, MNTB input width: p=0.0073).

The online version of this article includes the following source data and figure supplement(s) for figure 5:

**Source data 1.** Source data for *Figure 5A-D*.

**Figure supplement 1.** Spatial resolution of glutamate uncaging and additional examples of medial nucleus of the trapezoid body (MNTB) input maps recorded from wildtype and cKO mice.

**Figure supplement 1—source data 1.** Source data for *Figure 5—figure supplement 1*.

defined as the maximal distance of responsive uncaging sites along the mediolateral axis, was twice as large (*Figure 5A, B and D*; mean input width ± SEM: WT = 18% ± 3% of the MNTB's mediolateral length; cKO = 36% ± 5% of the MNTB's mediolateral length; p=0.0073, Student's *t*-test). The large increase in MNTB input width in cKO mice reveals that LSO neurons located in the medial high-frequency region of the nucleus not only received input from neurons of the high-frequency (medial) region of the MNTB, but also from neurons in the mid-nuclear region tuned to lower frequencies. For additional examples comparing the size and width of MNTB input areas between wildtype and cKO mice, see *Figure 5—figure supplement 1D and E*. These data strongly point toward an impairment of the tonotopic refinement of MNTB-to-LSO projections in cKO mice.

## Discussion

Kölliker's organ was identified as the origin of pre-sensory cochlear activity, which was suggested to serve the refinement of auditory circuits (*Tritsch et al., 2007*). The $Ca^{2+}$-activated $Cl^-$-channel TMEM16A, which is expressed in Kölliker's organ shortly before hearing onset, mediates spontaneous osmotic cell shrinkage of ISCs, which forces $K^+$ efflux and thus the transient depolarization of IHCs (*Wang et al., 2015*; *Yi et al., 2013*). Thereby, bursting activity of nearby IHCs, which will later respond to similar sound frequencies, becomes synchronized (*Eckrich et al., 2018*; *Harrus et al., 2018*; *Wang et al., 2015*), establishing a possible scenario for tonotopic map refinement (*Johnson et al., 2011*; *Tritsch and Bergles, 2010*; *Tritsch et al., 2007*). Here, we demonstrate that disruption of TMEM16A in the inner ear impairs prehearing cochlear activity and spontaneous burst firing as well as sensitivity and frequency selectivity of sound-evoked firing of MNTB neurons. Finally, we show that these alterations in prehearing auditory activity severely impair the refinement of the MNTB-LSO pathway.

Spontaneous activity of Kölliker's organ is characterized by recurrent ATP-dependent changes in cell volume and $Ca^{2+}$ transients of ISCs. The decreases in cell volume reflect the passive movement of water associated with $Cl^-$ efflux upon activation of the $Ca^{2+}$-activated $Cl^-$-channel TMEM16A (*Wang et al., 2015*; *Yi et al., 2013*). In WT mice, we detected spontaneous $Ca^{2+}$ waves and volume changes of ISCs as reported previously (*Tritsch et al., 2007*). In cKO mice, however, volume changes were absent and $Ca^{2+}$ waves were reduced to local $Ca^{2+}$ transients. Changes in cell volume can serve as a mechanical stimulus that activates robust and large ATP release (*Praetorius and Leipziger, 2009*) and, in turn, ATP release through connexin hemichannels is attributed to the propagation of $Ca^{2+}$ waves through the cochlear epithelium (*Anselmi et al., 2008*; *Schütz et al., 2010*; *Mammano, 2013*; *Jovanovic and Milenkovic, 2020*; *Mazzarda et al., 2020*). Therefore, TMEM16A-dependent volume changes may amplify ATP release via connexin hemichannels and thus promote the propagation of local $Ca^{2+}$ transients as $Ca^{2+}$ waves. ATP-mediated amplification of the $Ca^{2+}$ signals may further activate $Ca^{2+}$-activated TMEM16A $Cl^-$ channels in a positive feedback mechanism. First support of this model comes from our observation that the purinergic receptor blocker suramin abolished $Ca^{2+}$ waves in WT cochlea (as also shown by *Tritsch et al., 2007*) but did not affect the size of $Ca^{2+}$ transients in cKO mice.

In corneal endothelial cells, which express TMEM16A as well (*Cao et al., 2010*), mechanical stimuli can induce similar ATP- and connexin hemichannel-dependent $Ca^{2+}$ waves (*Gomes et al., 2005a*; *Gomes et al., 2005b*). Airway, submandibular gland, olfactory epitehlium and biliary epithelia also express TMEM16A and can generate ATP-release-dependent $Ca^{2+}$ waves following mechanical stimulation (*Hegg et al., 2009*; *Henriques et al., 2019*; *Dutta et al., 2013*; *Felix et al., 1996*; *Grygorczyk and Hanrahan, 1997*; *Homolya et al., 2000*; *Homolya et al., 1999*; *Romanenko et al., 2010*; *Ryu et al., 2010*; *Sanderson et al., 1990*; *Watt et al., 1998*). Thus, a positive feedback mechanism between TMEM16A and ATP receptors may also apply to other epithelia.

To analyze the impact of the disruption of TMEM16A on the auditory pathway, we performed in vivo juxtacellular recordings in the MNTB of cKO mice before and shortly after hearing onset. Burst-firing pattern of MNTB neurons was often completely absent or severely diminished in P9 cKO mice. MNTB neurons from P14 cKO mice also showed markedly diminished firing rates, indicating that they cannot respond to moderate-to-high-intensity stimulation with sufficient firing rates that are normally observed in wildtype littermates. As high sound intensities are coded by high firing rates, interaural-level differences might be affected and hence also sound source localization in cKO mice. Furthermore, individual MNTB neurons recorded from P14 cKO mice responded to sound stimuli from a much larger frequency range compared to wildtype mice. The reduced frequency selectivity could

result from superfluous functional projections from globular bushy cells (GBCs) onto MNTB neurons. At P2, MNTB neurons receive on average 9.3 inputs from GBCs, which after a short period of intense competition and pruning end up in one calyx of Held, usually by P9 (*Holcomb et al., 2013*). Because the elimination of excessive inputs is probably activity-dependent (*Holcomb et al., 2013*), reduced bursting activity in cKO mice might result in MNTB neurons that receive more than one input and thus show broadened FRAs.

To test whether projection patterns between auditory nuclei in cKO mice are indeed altered, we examined MNTB projections to the LSO, a part of the sound localization pathway in which refinement has been well studied (*Clause et al., 2014*; *Kim and Kandler, 2003*; *Müller et al., 2019*). The necessary spatial resolution was achieved with a fast galvanometric photolysis system, which allowed a fivefold improvement in area resolution compared to fiber optic-based uncaging systems used in previous studies (*Clause et al., 2014*; *Kim and Kandler, 2003*; *Kim and Kandler, 2010*). In cKO mice, the number of functional connections of single LSO afferents (MNTB input area) and the respective mediolaterally covered area, that is, the area along the tonotopic axis (MNTB input width), was twice as large as in wildtype. Since it has been reported that LSO neurons normally lose about 50% of their afferents during the first two weeks of postnatal development of the mouse (*Noh et al., 2010*), the present results imply that silencing of synaptic connections was strongly diminished in cKO mice.

Processes that underlie sensory network refinement are well studied in the visual system: Before onset of vision, bursting activity of retinal ganglion cells leads to the refinement of their projections to lateral geniculate nucleus neurons (*Wong et al., 1993*; *Penn et al., 1998*). This process depends on the duration of bursts, inter-burst intervals, and the synchronization of bursts (*Stellwagen and Shatz, 2002*; *Torborg et al., 2005*; *Butts et al., 2007*), and was proposed to follow Hebbian plasticity rules (*Hebb, 1949*). Similar mechanisms also apply for the auditory system, where neighboring IHCs show synchronized bursting activity before the onset of hearing (*Tritsch et al., 2007*). Notably, the firing patterns are less synchronized in cKO mice since simultaneous $K^+$ release is not triggered in the absence of TMEM16A-mediated $Cl^-$ efflux (*Wang et al., 2015*). Because synchronized bursting activity of neighboring IHCs (*Tritsch et al., 2007*) is faithfully relayed to the brainstem (*Tritsch et al., 2010*), it is likely that LSO neurons receive desynchronized MNTB inputs in cKO mice. In WT mice, GABA spillover from MNTB axon terminals can lead to the excitation of nearby synapses via pre- and extrasynaptic $GABA_A$ receptors (*Weisz et al., 2016*; *Fischer et al., 2019*) and thus likely contributes to the synchronization of MNTB inputs. In cKO mice, however, the shorter bursting will diminish GABA spillover, thus further compromising the synchronization of MNTB inputs. Hence, only in WT mice tonotopically similar axons that overlap in the LSO can be simultaneously active and get strengthened, possibly via glycinergic LTP (*Bach and Kandler, 2020*), whereas tonotopically distinct axons that have less overlap in the LSO get silenced following associative plasticity rules (*Hebb, 1949*). This assumption is in agreement with the recent discovery that synchronization of $Ca^{2+}$ signals of neighboring outer hair cells (OHCs) is required for proper refinement of afferent projections onto OHCs as well as the maturation of OHCs (*Ceriani et al., 2019*). The synchronization of OHC activity is also mediated by ATP release from supporting cells, namely, Deiter's cells, during the first two postnatal weeks in the mouse (*Jeng et al., 2020*).

Besides spontaneous activity of ISCs, IHC-firing patterns are also modulated by cholinergic medial olivocochlear efferents (*Glowatzki and Fuchs, 2000*; *Johnson et al., 2011*; *Sendin et al., 2014*), which transiently innervate immature IHCs (*Simmons et al., 1996a*; *Simmons et al., 1996b*; *Warr and Guinan, 1979*). In mice that lack the α9 subunit of nicotinic acetylcholine receptors (α9 KO mice), this cholinergic input is disrupted (*Clause et al., 2014*). The fact that α9 KO mice show more subtle changes in the bursting behavior of MNTB neurons in vivo (*Clause et al., 2014*) than our cKO mice suggests a prevalent role of the activity of ISCs in shaping firing patterns of the early auditory pathway prior to onset of hearing. Still, our results showed reduced synaptic refinement before hearing onset similar to those of α9 KO mice (*Clause et al., 2014*) and otoferlin KO mice, which exhibit drastically diminished spontaneous activity (*Müller et al., 2019*) due to disrupted glutamate release from IHCs (*Roux et al., 2006*). While otoferlin KO mice have almost no discernible bursting activity, cKO mice showed a drastic reduction of the number of bursts. In contrast, the number of bursts was not changed in α9 KO mice. Firing rates within bursts did not differ between cKO and WT mice, but were 70–80% higher in α9 KO compared to WT mice. Overall firing rates, however, do not differ between both KO models. Consequently, we infer that the overall firing rate and firing rates within bursts, as

well as the number of bursts, do not influence the physiological refinement of the MNTB-LSO pathway. Notably, the duration of bursts was markedly reduced in both KO mice (50% in α9 KO and 56% in cKO mice). Average ISIs, however, were significantly longer in cKO and shorter in α9 KO mice. Thus, it seems that both subtle and drastic changes in the temporal pattern and/or the duration of bursts can lead to a severe disruption of tonotopic maps. These changes have permanent consequences since tonotopic refinement of MNTB-LSO projections does not continue after hearing onset (*Müller et al., 2019*; *Walcher et al., 2011*). Thus, α9 knockout mice display diminished sound localization and sound frequency processing after hearing onset as revealed by behavioral studies (*Clause et al., 2017*).

Whether TMEM16A-mediated prehearing activity has a similar impact on hearing after hearing onset remains to be determined. While ABR measurements in our study did not reveal gross alterations in signaling along the lower auditory pathway, our recordings from individual auditory brainstem neurons showed significantly elevated sound thresholds. We note that the discrepancy between significantly indistinguishable ABR thresholds and significantly elevated auditory thresholds of single MNTB neurons could arise for several reasons. First, by the nature of population vs. single-neuron responses, a direct correspondence is not warranted. Secondly, we speculate that the poorer tonotopic refinement of the auditory circuitry might have contributed to the higher thresholds of MNTB neurons involving cochlear excitation from IHCs off the peak of the traveling wave. However, other reasons cannot be excluded and a trend toward higher ABR thresholds was apparent for 6 and 24 kHz tone bursts.

The elevated sound thresholds and frequency selectivity that we observed in cKO mice at P14 could indicate impaired auditory processing after hearing onset. This is supported by findings showing that mutations in ion channels (hemichannel, gap junctions, and $Ca^{2+}$ channels) involved in prehearing activity of Kölliker's organ are linked to congenital deafness (*Majumder et al., 2010*). Recent publications have established a causal connection between the bursting patterns of the ascending auditory system and the tonotopic refinement of MNTB projections to the LSO (*Clause et al., 2017*; *Müller et al., 2019*), and our work supports this notion. Moreover, our results link the propagation of $Ca^{2+}$ waves in developing supporting cells as a peripheral mechanism contributing to the refinement of ascending circuits.

# Materials and methods

**Key resources table**

| Reagent type (species) or resource | Designation | Source or reference | Identifiers | Additional information |
|---|---|---|---|---|
| Gene (mouse) | *Ano1* (*Tmem16a*) | GenBank | MGI:2142149 | |
| Strain, strain background (*Mus musculus*) | *Ano1*fl/fl | Published in *Heinze et al., 2014* | | |
| Strain, strain background (*M. musculus*) | *Ano2* KO | Published in *Billig et al., 2011* | | |
| Strain, strain background (*M. musculus*) | Pax2Cre | Published in *Ohyama and Groves, 2004* | | |
| Strain, strain background (*M. musculus*) | C57BL6/J | Jackson | 000664 | |
| Antibody | Anti-TMEM16A polyclonal rabbit | Published in *Heinze et al., 2014* | | IF (1:500) |
| Antibody | Anti-TMEM16B polyclonal rabbit | Published in *Billig et al., 2011* | | IF (1:100) WB (1:1000) |
| Antibody | Anti-CD31 monoclonal rat | BioLegend | 102,502 | IF (1:500) |
| Antibody | Alexa Fluor 488, polyclonal goat anti-rabbit | Molecular Probes | A11008 | IF (1:1000) |
| Antibody | Cy3, polyclonal donkey anti-rabbit | Jackson ImmunoResearch | 711-165-152 | IF (1:10,000) |

*Continued on next page*

*Continued*

| Reagent type (species) or resource | Designation | Source or reference | Identifiers | Additional information |
|---|---|---|---|---|
| Antibody | DyLight 488 polyclonal goat anti-rat | Bethyl Laboratories | A110-305D2 | IF (1:1000) |
| Antibody | HRP polyclonal goat anti-guinea pig | Merck | AP108P | WB (1:1000) |
| Chemical compound, drug | Fura-2 AM | Thermo Fisher | F1221 | $Ca^{2+}$ dye |
| Chemical compound, drug | MNI-caged glutamate trifluoroacetate | Femtonics | | Caged glutamate |
| Chemical compound, drug | DAPI stain | Invitrogen | D1306 | (1 μg/ml) |
| Chemical compound, drug | Fluorogold hydroxystilbamidin-bis-(methansulfonat) | Sigma-Aldrich | 39286 | Fluorescent tracer |
| Chemical compound, drug | Ketavet Ketamine hydrochloride | Pfizer | | 0.1 mg/g body weight |
| Chemical compound, drug | Rompun Xylazine hydrochloride | Bayer | | 0.5 μg/g body weight |
| Software, algorithm | SigmaPlot 12.5 | Systat Software Inc | | Statistical analysis |
| Software, algorithm | GraphPad Prism | GraphPad Software Inc | | Statistical analysis |

## Mouse strains

Mouse care and usage were in accordance with the German animal protection laws and were approved by the local authorities (license numbers: 33.9-42502-04-11/0439; TVV 06/09 and TLV UKJ-17-006). *Ano1*[fl/fl] mice were crossed with a *Pax2*[Cre] mouse line (a gift from A. Groves, House Ear Institute, Los Angeles) to obtain *Ano1* ear conditional knockout mice. Throughout the article, these mice are referred to as cKO mice. Information about the mouse strains including genotyping primers can be found in *Heinze et al., 2014* and *Ohyama and Groves, 2004*. *Ano2* knockout mice were a gift from T. Jentsch, Leibniz Institute for Molecular Pharmacology (Germany), and were described in *Billig et al., 2011*.

## Immunohistochemistry

Cochlear cryosections were prepared from mice at postnatal ages P0, P3, P9, P13, and P15. Mice were decapitated and inner ears (cochlea capsule attached to the vestibular apparatus) were isolated and processed according to a protocol from *Whitlon et al., 2001*. In brief, inner ears were fixed at room temperature (RT) in 4% paraformaldehyde (PFA) in phosphate-buffered saline (PBS) for 11.5 hr. Inner ears from mice older than P5 were decalcified in 10% EDTA (PBS) at 4°C for 12 hr to up to 48 hr depending on the age of the mouse. Inner ears were cryoprotected with solutions with increasing sucrose concentrations (10–30% sucrose [PBS]) and kept in tissue Tek (Sakura) at 4°C. Embedded inner ears were shock frozen using a dry ice/ethanol bath. Cryosections (10 μm) of inner ears were cut in the coronal plane using a cryostat (Leica CM3050S cryostat). Whole-mount brain slices were prepared from mice at P8. Coronal slices (150 μm) were cut on a vibrating microtome (Leica VT1200S) in dissection solution containing (in mM) 50 NaCl, 2.5 KCl, 1.25 $NaH_2PO_4$, 3 $MgSO_4$, 0.1 $CaCl_2$, 75 sucrose, 25 D-glucose, 25 $NaHCO_3$, and 2 Na pyruvate (280 mOsmol, pH 7.2).

Cochlea cryosections and brain slices were fixed in 4% PFA in PBS at RT for 20 min and incubated in blocking solution containing 1% fetal bovine serum (BSA), 3% goat serum (with 0.02% $NaN_3$) (Invitrogen), 3% donkey serum (Millipore), and 0.5% Triton X100 in PBS at RT for 3 hr. To detect TMEM16A and TMEM16B, cryosections and/or brain slices were incubated at 4°C for 24 hr in primary antibody solution containing blocking solution and a rabbit polyclonal antibody directed against TMEM16A (1:500) (*Heinze et al., 2014*) and a guinea pig polyclonal antibody directed against TMEM16B (1:100) (*Billig et al., 2011*). To visualize blood vessels in brain slices, sections were co-stained with a primary monoclonal rat antibody against CD31 (1:500, BioLegend [102502]). Cryosections of the cochlea and brain slices were washed in PBS for 1 hr and subsequently incubated in secondary antibody solution containing blocking solution and the secondary antibodies Alexa Fluor 488 conjugated goat anti-rabbit (1:1000, cochlear cryosections) (Molecular Probes [A11008]), Cy3 conjugated donkey anti-rabbit

(1:1000, brain slices) (Jackson ImmunoResearch [711-165-152]), and DyLight 488 conjugated goat anti-rat (1:1000, brain slices) (Bethyl Laboratories [A110-305D2]) at 4°C for more than 12 hr. Cochlear cryosections and brain slices were washed with PBS for 1 hr, counterstained with DAPI to visualize cell nuclei, and mounted on microscope slides (ProLong Gold anti-fade reagent, Life Technologies). Images were captured using an upright confocal microscope (LSM 710, Zeiss).

### Hematoxylin-eosin staining

Cochlear cryosections (see 'Immunohistochemistry') were fixed in 4% PFA (in PBS) at RT for 20 min. For hematoxylin-eosin (HE) staining, the Robot-Stainer device from Microm (HMS 740) was used.

### Western blot

Modioli including the SGNs were isolated from *Ano2* knockout mice and their wildtype littermates (P5). For immunoblots, the tank blotting Mini-Protean Tetra system (Bio-Rad) and transfer buffer with methanol (25 mM Tris, 192 mM glycine) were used. 10 µg of proteins were separated in 5% acrylamide gels. Proteins were blotted on polyvinylidene membranes (Roti-PVDF, Roth). Protein transfer was controlled by staining with Ponceau S (0.2% Ponceau S, 3% acetic acid). Blocking was done in Tris buffered saline (TBS) supplemented with 0.05% Tween-20 and 5% dry milk powder. TMEM16B was detected using the same antibody as described in the immunohistochemistry section (1:1000). It was incubated overnight at 4°C in TBS-T with 1% dry milk powder. Washing was done with TBS/0.05% Tween-20. The secondary antibody (goat anti-guinea pig IgG antibody-HRP, Merck, 1:1000) was incubated at RT for 2 hr in TBS-T with 1% dry milk powder. Signals were visualized by chemiluminescence (Amersham ECL Prime Western Blotting Detection Reagent, GE Healthcare). Documentation was done with the camera system Stella 3200 (Raytest) and the Xstella software.

### DIC/Ca$^{2+}$ imaging

After isolation of the inner ear from 5-to-7-day-old mice, the cochlea was removed from the capsule and the spiral ligament was dissected away. The cochlea was then mounted on a poly-L-lysine-coated (100 µg/ml) Petri dish. Dissections and imaging were done in artificial cerebrospinal fluid (ACSF) containing (in mM) 130 NaCl, 3.5 KCl, 2 CaCl$_2$, 1.3 MgCl$_2$, 1.2 NaHPO$_4$, 25 NaHCO$_3$, and 25 D-glucose (320 mOsmol, pH 7.4). The ACSF was saturated with 95% O$_2$/5% CO$_2$.

For DIC imaging, acutely isolated cochleae were transferred to an inverted microscope (Axio Observer.Z1, Zeiss). A time-lapse image series was taken for 20 min with a ×25 water-immersion objective. One image per second was acquired (1392 × 1040 px) using a CCD camera (Photometrics Cool Snap HQ$^2$, Visitron) and the Metamorph imaging software (Visitron).

Ca$^{2+}$ signals were visualized by Ca$^{2+}$ imaging of cochleae that had been bath-loaded with 10 µM Fura-2 AM (Invitrogen) and 0.02% pluronic acid F-127 (Invitrogen) in ACSF at RT for 35–45 min. Loading was followed by another 30 min incubation in ACSF. Cochleae were imaged on an upright microscope (Axio Observer.Z1, Zeiss) using a ×25 water-immersion objective. Using the MetaFluor imaging software (Visitron), 400 time-lapse ratio images (1392 × 1040 px) were obtained at a rate of one image per second. Therefore, pairs of images were acquired at alternate excitation wavelengths (340/380 nm). The emission was filtered at 500–530nm.

Images acquired during DIC- and Ca$^{2+}$ imaging were processed using a custom-written ImageJ macro and the ImageJ software (NIH) deposited at https://github.com/alenameis/ANO1-hearing-development.git (copy archived at swh:1:rev:4af9cb8bc927320653b556eab12000e675a61e12; *Maul, 2022*). Each frame (time point $t_n$) was subtracted from an average of five preceding frames: $t_n - [\frac{t_{n-1}+t_{n-2}+t_{n-3}+t_{n-4}+t_{n-5}}{5}]$. The mean areas and frequencies of volume changes and Ca$^{2+}$ events were analyzed within a 10,000 µm$^2$ area of Kölliker's organ that was chosen from the center of the field of view. The frequencies of volume changes and Ca$^{2+}$ events were analyzed by counting the number of events per second. The area of each event was measured in square micrometer using the Metamorph or MetaFluor imaging software for volume changes or Ca$^{2+}$ signals, respectively.

To be regarded as an event, volume changes had to match the following two criteria: first, events had to show an increase in DIC contrast above 10% of the baseline level. Baseline contrast levels were measured from an area of the cochlea that did not show volume changes (i.e., an area outside of Kölliker's organ and the IHC region). Second, areas with changes in DIC contrast had to be larger than the surface area of one ISC.

To be regarded as an event, $Ca^{2+}$ signals had to match the following two criteria: first, events had to show an increase in fluorescence intensity greater than 10% of the baseline level. Baseline fluorescence levels were measured from an area of the cochlea that did not show $Ca^{2+}$ events (i.e., an area outside of Kölliker's organ and the IHC region). Second, areas with changes in fluorescence had to be larger than the surface area of one ISC. Videos were generated with the ImageJ software (seven images per second) and annotated using Premiere Pro (Adobe).

## In vitro electrophysiological recording and functional mapping

The brain was removed from mice (P9–11) and the forebrain and parts of the cerebellum were dissected to isolate the brainstem. Coronal brainstem slices (300–400 μm) were cut on a vibrating microtome (Leica VT1200S) in ice-cold dissection solution containing (in mM) 50 NaCl, 2.5 KCl, 1.25 $NaH_2PO_4$, 3 $MgSO_4$, 0.1 $CaCl_2$, 75 sucrose, 25 D-glucose, 25 $NaHCO_3$, and 2 Na pyruvate (280 mOsmol, pH 7.2). The solution was oxygenated with 95% $O_2$ and 5% $CO_2$ during dissection. A single brain slice per animal was obtained, which contained the MNTB and LSO to reach a maximum possible preservation of connections between these two nuclei. Slices recovered for 30 min at 36°C and for an additional 30 min at RT in oxygenated recording solution containing (in mM) 125 NaCl, 2.5 KCl, 1.25 $NaH_2PO_4$, 2 $MgSO_4$, 2.5 $CaCl_2$, 18 D-glucose, and 25 $NaHCO_3$ (290 mOsmol, pH 7.2). For electrophysiological recordings, brain slices were transferred to a recording chamber, mounted on an upright microscope (Axio Examiner A.1, Zeiss), and perfused with oxygenated recording solution at a speed of 2 ml/min using a pressure-driven perfusion system (ALA-VM8, ALA Instruments). The LSO was identified using a digital camera (C10600 Orca $R^2$, Hamamatsu). Recordings were made from neurons with bipolar morphology from the medial (high frequency) region of the LSO in order to obtain comparable results and because the highest degree of refinement was reported for this frequency region (Sanes and Siverls, 1991). Electrodes had a resistance of 4–5 MΩ (Biomedical Instruments) when filled with intracellular solution containing (in mM) 54 potassium gluconate, 56 KCl, 1 $MgCl_2$, 1 $CaCl_2$, 5 sodium phosphocreatine, 10 HEPES, 11 EGTA, 0.3 NaGTP, and 2 MgATP (285 mOsmol, pH 7.2). Recordings were done in current-clamp in the whole-cell configuration at RT, and LSO neurons were held at 70 mV. Currents were acquired with an Axon CNS MultiClamp 700B amplifier (Molecular Devices) and lowpass filtered (3 kHz) and digitized (3 kHz) with a Digidata 1440A analog-to-digital converter (Molecular Devices) using the pClamp10 software (Molecular Devices).

## Photolysis of caged glutamate

Presynaptic MNTB neurons were localized by local uncaging of 120 μM MNI-caged glutamate trifluoroacetate (Femtonics) that was added to the recording solution. Uncaging of glutamate was controlled by a fast galvanometric photolysis system (UGA40, Rapp OptoElectronic) that was triggered via a TTL impulse. A 405 nm continuous diode laser (DL405, Rapp OptoElectronic) was used as a light source. Laser flashes were delivered through a light guide of 20 μm diameter (Rapp OptoElectronic) with a ×10 water-immersion objective that produced circular spots of 20 μm diameter in the focal plane. For mapping, a virtual grid was defined over (and around) the MNTB harboring 250,300 grid points (spaced 20 μm apart) using the UGA-40 control 1.1 software (Rapp OptoElectronic). Each grid point represented one uncaging site. Laser pulses of 10 ms duration were delivered with 6 s delay time between uncaging sites. This resulted in reproducible LSO-MNTB input maps confirmed by rescanning in some cases. Only one MNTB map was obtained per mouse.

## Reconstruction and analysis of MNTB-LSO input maps

Events were detected with template search methods during a 20 ms window starting from the onset of the laser illumination. Peak amplitudes of the postsynaptic potentials (PSPs) were measured (Clampfit 10.2 software, Molecular Devices). Each uncaging site that evoked PSP amplitudes greater than 2 mV in the recorded LSO neuron was considered as a functional MNTB-LSO connection and marked as colored square according to the following color code: yellow = depolarizations > 2 mV, orange > 10 mV, red = action potential. Stimulation sites that evoked PSPs smaller than 2 mV or no PSPs were considered as unresponsive MNTB areas and were left blank. The total area covered by all colored squares was defined as the MNTB input area. The analysis of the MNTB input area was done in pixels (1 pixel represents 1 μm²). The MNTB input area was calculated by multiplying the size of an uncaging site (~400 px) by the number of colored squares. The MNTB input area was normalized to

the cross-sectional area of the MNTB. MNTB borders were determined by two investigators (one of them was blind to the mouse group) using images taken from the slices during the experiment. The result represents the MNTB input area in percent. The MNTB input width was calculated by measuring the maximal distance of stimulations sites that evoked depolarizations greater than 10 mV along the mediolateral axis. This distance was normalized to the mediolateral length of each MNTB. The result expresses the input width in percent along the tonotopic axis of the MNTB. On rare occasions, responses could be elicited from uncaging sites slightly ventral or dorsal to the defined MNTB boundaries in mice of both genotypes. These sites were included in the analysis.

## Spatial resolution of glutamate uncaging

The spatial resolution was assessed by direct current-clamp whole-cell recordings from MNTB neurons. The holding potential was –70 mV and the electrode resistance was 34 MΩ when filled with intracellular solution containing (in mM) 54 potassium gluconate, 56 KCl, 1 $MgCl_2$, 1 $CaCl_2$, 5 sodium phosphocreatine, 10 HEPES, 11 EGTA, 0.3 NaGTP, and 2 MgATP (285 mOsmol, pH 7.2). A virtual 10 × 10 grid was defined over the recorded MNTB neuron with uncaging points spaced 10 μm apart. The 'action potential-eliciting distance' was defined as the maximal distance from the center of the uncaging site to the center of the cell body at which uncaging produced action potentials in the recorded MNTB neuron. The action potential-eliciting distance was measured for the mediolateral and dorsoventral direction of the MNTB. We used the determined action potential-eliciting distance to define the uncaging parameters for the mapping experiment.

## In vivo recordings in the MNTB

Juxtacellular single-unit recordings were performed in mice before or right after hearing onset (P8, P14). Animals were anesthetized with an initial intraperitoneal injection of a mixture of ketamine hydrochloride (0.1 mg/g body weight; Ketavet, Pfizer) and xylazine hydrochloride (5 μg/g body weight; Rompun, Bayer). Throughout recording sessions, anesthesia was maintained by additional subcutaneous application of one-third of the initial dose, approximately every 90 min. The MNTB was approached dorsally and typically reached at penetrations depths of 5000–5500 μm.

The experimental protocol for acoustic stimulation and single-unit recording was described in detail previously (*Dietz et al., 2012*; *Sonntag et al., 2009*). Briefly, auditory stimuli were digitally generated using custom-written MATLAB software (The MathWorks Inc, Natick) at a sampling rate of 97.7 kHz. The stimuli were transferred to a real-time processor (RP2.1, Tucker-Davis Technologies), D/A converted and delivered through custom-made earphones (acoustic transducer: DT 770 pro, Beyer Dynamics). Two recording protocols were used: (i) pure tone pulses (100 ms duration, 5 ms rise-fall time, 100 ms interstimulus interval) were presented within a predefined matrix of frequency/ intensity pairs to determine the excitatory response areas of single units. CF (sound frequency causing a significant increase of unit's action potential spiking at the lowest intensity), the respective threshold, Q-values ($Q_n$, a measure of the unit's sharpness of tuning calculated as the ratio of CF to bandwidth at n = 10, 20 and 30 dB above threshold), and maximum discharge rates were computed from the response area and used for the next protocol. (ii) Spontaneous neuronal discharge activity was assessed in the absence of acoustic stimulation to determine the average firing rate and CV of ISIs (ISIs = time that passes between two successive action potentials). Regularity of spontaneous discharge activity was quantified by CV, calculated as the ratio of standard deviation of ISIs and mean ISI. Additional analysis was done for MNTB units recorded in P8 mice, where spontaneous spiking discharges are grouped in bursts followed by periods of greatly reduced discharge activity (*Sonntag et al., 2009*). We used a statistical test based on gamma probability distribution of ISIs to determine the number and duration of bursts, and number of spikes per burst within each single-cell recording.

For the juxtacellular single-unit recordings, glass micropipettes (GB150F-10, Science Products) were pulled on a horizontal puller (DMZ universal puller, Zeitz) to have a resistance of 5–10 M when filled with 3 M KCl. The recorded voltage signals were amplified (Neuroprobe 1600, A-M Systems), bandpass filtered (0.3–7 kHz), digitized at a sampling rate of 97.7 kHz (RP2.1, Tucker-Davis Technologies), and stored for offline analysis using custom-written MATLAB software. Three criteria were used to classify single-unit recordings: (i) changes in the spike height did not exceed 20%, (ii) uniform waveforms, and (iii) signal-to-noise ratio at least 8:1. The principal neurons of the MNTB were identified by the complex waveform of the recorded discharges (*Guinan and Li, 1990*; *Sonntag et al., 2009*).

Histological verification of the recording site was done by iontophoretic injection of Fluorogold (4 µA for 7 min). The animal was perfused 4–6 hr after injection with 0.9% NaCl solution followed by 5% PFA. The brain was cut on a vibratome (Microm HM650), and the tissue sections (100 µm thick) were visualized under the fluorescent microscope (Axioskop, Zeiss). An example of a recording site is shown in *Figure 2I*.

## Auditory brainstem response

ABR recordings were conducted as described previously (*Jing et al., 2013*). In brief, mice (P13–14) were anesthetized intraperitoneally with a combination of ketamine (125 mg/kg) and xylazine (2.5 mg/ kg). The ECG was constantly monitored to control the depth of anesthesia. The core temperature was maintained constant at 37°C using a temperature-controlled heat blanket (Hugo Sachs Elektronik– Harvard Apparatus). Note that in these small immature mice the temperature probe could not be placed rectally, contributing to the relatively long latencies and greater variability of the ABR waves. ABR peaks IV (~5.3 ms) and V (~7 ms) were very small and were thus excluded from analysis. For stimulus generation, presentation, and data acquisition, the TDT System II (Tucker-Davis Technologies) was used that was run by the BioSig32 software (Tucker-Davis Technologies). SPLs were provided in dB SPL root mean square (RMS) (tonal stimuli) or dB SPL peak equivalent (clicks) and were calibrated using a 1/4 inch Brüel and Kjær microphone (model 4939). Tone bursts (6/12/24 kHz, 10 ms plateau, 1 ms cos$^2$ rise/fall) or clicks of 0.03 ms were presented at 40 Hz or 20 Hz, respectively, in the free field ipsilaterally using a JBL 2402 speaker. The difference potential between vertex and mastoid subdermal needles was amplified (50,000×) and filtered (400–4000 Hz, NeuroAmp) and sampled at a rate of 50 kHz for 20 ms, 1300× to obtain two mean ABRs for each sound intensity. Hearing threshold was determined with 10 dB precision as the lowest stimulus intensity that evoked a reproducible response waveform in both traces by visual inspection.

## Statistics and statistical significance

For statistical analysis, the SigmaPlot 12.5 (Systat Software Inc) and GraphPad Prism software were used. Datasets with normal (Gaussian) distribution were assessed by unpaired Student's *t*-tests (two-tailed distribution) if not said otherwise. For datasets with non-Gaussian distribution, the nonparametric Mann–Whitney rank-sum test was used. For analysis of the intensity dependence of ABR amplitudes and latencies and of ABR thresholds across frequencies, two-way ANOVA was used. If the data were normally distributed, the results are displayed as means ± standard error of means (s.e.m.) and otherwise as medians with the respective 25% and 75% quartiles. Mean cumulative distributions of interevent intervals and distribution of ISIs between wildtype and cKO groups were compared using the two-sample Kolmogorov–Smirnov test and the chi-square test, respectively (see *Figure 2*). To avoid assigning too much weight on high rate-spiking cells, distributions were created by pooling n random ISIs for each cell within the two groups (n = lowest number of events recorded in any of the cells). A statistical test based on gamma probability distribution of ISIs was used to determine the number and duration of bursts and number of spikes per burst within each single-cell recording. In brief, we assumed that action potential firing of MNTB neurons is a random Poisson-like process. In this way, the probability to encounter one ISI in time period $\tau$ is p=1-e$^{-\lambda \tau}$, where $\lambda$ stands for neuronal firing rate. Next, the probability to encounter *k* ISIs during the time period $\tau$ as *k*-fold convolution of exponential distribution density was calculated, thereby yielding gamma distribution of waiting times for *k* ISIs. The resulting probability is $p = \int_0^{\tau} x^{k-1} \frac{\lambda^k e^{-\lambda s}}{(k-1)!} dx$. We ran the statistical test through our recorded action potential times, and each spike train where probability stayed lower than 0.01 (p< 0.01) for at least 10 ISIs (k ≥ 10) was recognized as a burst. Significance levels < 0.05 are denoted by *, <0.01 **, and <0.001 ***. No statistical methods were used to predetermine sample sizes.

## Acknowledgements

We thank Andrew Groves for the opportunity to use the Pax2Cre mouse line. We further acknowledge the technical support provided by Anje Sporbert and Zoltan Cseresnyes from the microscope core facility of the Max Delbrück Center for Molecular Medicine in the Helmholtz Association. This study was funded by a grant of the Thyssen foundation to Björn Schroeder and CAH, by grants of

the BMBF (01EW1706) and the DFG to CAH (HU 800/10-1) and grants of the DFG to NS, RR, and TM (priority program 1608). In addition, the work of TM was supported by Fondation Pour l'Audition (FPA RD-2020-10).

## Additional information

### Funding

| Funder | Grant reference number | Author |
|---|---|---|
| Deutsche Forschungsgemeinschaft | HU 800/10-1 | Christian A Hübner |
| Deutsche Forschungsgemeinschaft | priority program 1608 | Nicola Strenzke<br>Tobias Moser<br>Rudolf Rübsamen |
| Bundesministerium für Bildung und Forschung | 01EW1706 | Christian A Hübner |

The funders had no role in study design, data collection and interpretation, or the decision to submit the work for publication.

### Author contributions

Alena Maul, Conceptualization, Data curation, Formal analysis, Investigation, Methodology, Writing – original draft, Writing – review and editing; Antje Kathrin Huebner, Conceptualization, Investigation, Resources, Writing – review and editing; Nicola Strenzke, Data curation, Investigation, Writing – original draft; Tobias Moser, Rudolf Rübsamen, Conceptualization, Writing – review and editing; Saša Jovanovic, Data curation, Formal analysis, Investigation, Methodology, Writing – original draft, Writing – review and editing; Christian A Hübner, Conceptualization, Funding acquisition, Project administration, Resources, Supervision, Writing – original draft, Writing – review and editing

### Author ORCIDs

Nicola Strenzke ⬥ http://orcid.org/0000-0003-1673-1046
Tobias Moser ⬥ http://orcid.org/0000-0001-7145-0533
Christian A Hübner ⬥ http://orcid.org/0000-0002-1030-4943

### Ethics

This study was performed in strict accordance with the recommendations in the Guide for the Care and Use of Laboratory Animals of the National Institutes of Health. All of the animals were handled according to our local authorities (license numbers: 33.9-42502-04-11/0439; TVV 06/09 and TLV UKJ-17-006).

### Decision letter and Author response

Decision letter https://doi.org/10.7554/eLife.72251.sa1
Author response https://doi.org/10.7554/eLife.72251.sa2

## Additional files

### Supplementary files

• Supplementary file 1. Distribution of interspike intervals and quantification of auditory brainstem response. (a) Comparison of the distribution of interspike intervals (ISIs) between wildtype and cKO mice. (b) Quantification of auditory brainstem response (ABR) thresholds (mean ± SEM) in response to stimulation with tone bursts at 6, 12, and 24 kHz, or click stimulation. (c) Quantification of peak amplitudes (mean ± SEM) of the first three ABR waves (I–III) in response to click stimuli of various intensities (40–100 dB). (d) Quantification of latencies (mean ± SEM) of the first three ABR waves (I–III) in response to click stimuli of various intensities (40–100 dB).

• Transparent reporting form

## Data availability

All data generated or analysed during this study are included in the manuscript and supporting file; Source Data files have been provided and did not change for the revised manuscript.

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
