## [Editor Report]

This study addresses the extremely interesting question of how spontaneous activity in the cochlea prior to hearing onset impacts the development of auditory circuits in the brainstem. The study has many strengths, including the use of complementary in vitro and in vivo recording techniques to characterize both peripheral and central defects resulting from conditional deletion of the gene for the chloride channel TMEM16A. The reviewers identified some concerns over the interpretation of the data, but all of these concerns were addressed in the subsequent revisions.

---

## [Decision Letter]

**Decision letter after peer review:**

Thank you for submitting your article "The Cl^-^-channel TMEM16A controls the generation of cochlear Ca^2+^ waves and promotes the refinement of auditory brainstem networks" for consideration by *eLife*. Your article has been reviewed by 3 peer reviewers, one of whom is a member of our Board of Reviewing Editors, and the evaluation has been overseen by Andrew King as the Senior Editor. The reviewers have opted to remain anonymous.

Essential revisions:

All of the reviewers feel the results represent a significant advance. However the context in which many of the results was provided was not complete. This is the basis of most of the comments.

1) The author recapitulate the impact of the TMEM KO on spontaneous activity in the cochlear nucleus as reported in Wang et al., which all the reviewers feel is an important validation. However, the authors conclude that TMEM16A is upstream of ATP release from inner supporting cells, which is the opposite of which has been proposed by the previous study. There are no independent experiments supporting this claim – so either additional experiments need to be done, or the appropriate caveats to this model needs to be addressed. The authors are encouraged to read the detailed comments of Reviewer 3 on this matter.

2) The evidence that the data supports a claim of "axonal refinement" is perhaps overstated given the lack of anatomical evidence. Either the authors should provide additional data on this (Reviewer #1 comments) or more explicitly compare their results to the nAChR alpha9 KO study from the Kandler lab, which coupled similar uncaging methods with axon refinement (Reviewer #3 comments). Note the suggestions from Reviewer 3 to address this issue do not require additional experiments.

3) Although there are many suggestions from all three Reviewers, the authors are particularly encouraged to address the many points made by Reviewer #3.

*Reviewer #1 (Recommendations for the authors):*

The biggest question has to do with the overall impact of these findings and whether they represent a significant advance.

1. The authors need to clarify whether their results in Figure 1 (the impact of KO on spontaneous activity) contain novel findings or do they recapitulate the results of the Bergles lab using whole animal KOs. Specifically, they do not provide evidence that the TMEM16 KO is upstream of ATP release, as they conclude.

2. The conclusion that this disruption of activity leads to reduction in refinement in the early auditory pathway is based entirely on physiological measurements and therefore no change in axonal projection is directly observed. It appears there might be other interpretations of their findings. I will focus primarily on the mapping of MNTB inputs to the LSO, which was achieved by glutamate uncaging – I think the configuration is making a slice that contains both the MNTB and the LSO (it would be great for authors to clarify this). Then a neuron in LSO is recorded in current clamp while glutamate uncaging occurs over the MNTB, driving individual neurons to fire action potentials. They find that MNTB neurons from a larger area all synapse onto a single LSO neuron, indicating a lack of refinement. That is one interpretation of these findings but there are many others, such as changes in local circuitry. The authors would need to do more experiments to reach this conclusion. For example, they could do experiments in voltage clamp to assay impact of excitatory vs inhibitory inputs to assay which is more responsible for membrane potential changes. Second they should use these method to demonstrate that they see a change in this distribution across development to make clear that this refinement occurs at the same ages that spontaneous activity is disrupted. Finally, the authors need to provide axon reconstructions to support the idea of the map refinement.

*Reviewer #2 (Recommendations for the authors):*

This is a very well written and beautiful study.

*Reviewer #3 (Recommendations for the authors):*

This is a well written, and well-presented paper. As described in the "Public Review" portion, most weaknesses are limited to interpretations of results and fitting the work more clearly into the existing literature, so as to not over-state the novelty of the work.

Details:

– In the abstract, the final sentence does not fit with existing literature and is not supported by the presented data. Previous papers from the Bergles group (Wang et al. 2015, Cell) indicate that purine release from cochlear inner supporting cells is upstream of TMEM16A. This line of the abstract, as well as lines 165-166, state the opposite. However, your data does not adequately disprove the Bergles lab hypothesis and does not address which part of the pathway initiates the calcium wave process. Omit these lines or provide stronger evidence. This will also include re-writing the title of the paper, as there is no evidence that TMEM16A itself initiates the wave patterns, although it is one molecule in the chain.

– In figure 1 what was the total amount of time used for statistics? The 6 seconds of data shown is an extremely short time window, and not adequate to capture the variable timescales of the calcium waves.

– Line 113 – an extra space in the p-value.

– Results section beginning line 154: Tritsch et al. 2007 reports the effect of suramin in reducing crenellations in inner supporting cells. In the same paper the crenellations were associated with calcium waves. Therefore, the experiment reported here is extremely similar to previously published work and this other work should be cited.

– Figure 2 fonts are too small.

– Please discuss how the changes in burst patterns detailed in figure 2 compare to previously published studies linking aberrant cochlear spontaneous activity with impaired MNTB-LSO tonotopic projections that use nearly identical techniques (Clause et al. 2014).

– Line 207 – spell out approximately.

– MNTB neurons have higher thresholds, but ABR studies show normal thresholds. Please speculate as to why.

– Lines 283-284 would benefit from a very quick statistical test to demonstrate no difference between the CFs of MNTB neurons in the mutant and control groups.

– Re-write the sentence in lines 301-302 – perhaps move the word 'above' to after 'SPL'.

– Lines 306-307 – detail how many animals.

– Lines 307-309 appear to be totally speculative, if the statement remains it should be directly tested by comparing the firing rates of mutant and control MNTB neurons at the loudest sound levels tested.

– Lines 323-324 – briefly mention the illumination method here or in figure S6 legend.

– Line 540 – I am guessing that you are actually using an upright, not an inverted microscope, if it has a dipping objective.

– Lines 554-556 are confusing and need to be re-worded.

– Line 604 – please justify why the 2mV cutoff for a successful PSP was arbitrarily chosen. How does this relate to noise levels in the baseline?

– Was there any change in the total size of the MNTB in TMEM16A KO animals?

– Figure S6 – please add labeling to make the recording location (MNTB vs LSO neuron) more clear in the different panels.

[Editors' note: further revisions were suggested prior to acceptance, as described below.]

Thank you for resubmitting your work entitled "The Cl^-^-channel TMEM16A is involved in the generation of cochlear Ca^2+^ waves and promotes the refinement of auditory brainstem networks in mice" for further consideration by *eLife*. Your revised article has been reviewed by 3 peer reviewers, one of whom is a member of our Board of Reviewing Editors, and the evaluation has been overseen by Andrew King as the Senior Editor.

The manuscript has been improved but there are some remaining issues that need to be addressed, as outlined below:

For the most part, all reviewers felt that their concerns had been adequately addressed in the revision. There remains one issue regarding a point raised by Reviewers 1 and 3, and that has to do with the conclusions the authors reach regarding the mechanisms for the propagating calcium waves in ISCs. The authors conclude there is a "complex bi-directional interplay of TMEM16A and ATP receptors", but again the evidence the authors provide that TMEM16A influences ATP release is not apparent. Rather than there being "bidirectional signaling", the reviewers suggested that the authors should state that there is a positive feedback mechanism.

The new text lines 382-404 is perhaps more confusing – the confusion coming from mixing in the result from other epithelial cells with what is known about ISCs in developing cochlea. A revision of this paragraph would help clarify models in the minds of non-experts.

*Reviewer #1 (Recommendations for the authors):*

The authors have addressed my concerns by weakening their interpretations.

First, regarding the mechanisms for the propagating calcium waves in ISCs – the authors conclude "complex bi-directional interplay of TMEM16A and ATP receptors" but again the evidence it influences ATP release is limited – except through positive feedback mechanisms. Previous evidence for propagating waves in ISC (e.g. from the Bergles lab) should be stated more clearly.

Second, the only change made regarding the receptive field refinement is to call it "physiological refinement" with no discussion of potential mechanisms. Please clarify how you think frequency selectivity is altered.

*Reviewer #2 (Recommendations for the authors):*

The authors have answered all of my questions. The manuscript can be accepted for publication.

*Reviewer #3 (Recommendations for the authors):*

The authors have adequately addressed my concerns.

---

## [Author Response]

Reviewer #1 (Recommendations for the authors):The biggest question has to do with the overall impact of these findings and whether they represent a significant advance.1. The authors need to clarify whether their results in Figure 1 (the impact of KO on spontaneous activity) contain novel findings or do they recapitulate the results of the Bergles lab using whole animal KOs. Specifically, they do not provide evidence that the TMEM16 KO is upstream of ATP release, as they conclude.

We would like to stress that in contrast to the publication by Wang et al., which reports local Ca^2+^ transients and SC crenations in control cochleae, we here show that these events propagate as waves along the cochlea and that these coordinated activity is absent in the cKO. So we add another important piece of information to the field beyond the data provided by Wang et al..

Moreover, we show that the inhibition of purinergic receptors with suramin abolishes the propagation of Ca^2+^ transients as waves. This finding is not new on its own but in connection with the data provided by Wang et al. and the fact that the ATP-dependent propagation of Ca^2+^ waves was abolished in an organ-on-chip recordings of ISCs devoid of connexin30 (Mazzarda, D'Elia et al. 2020), it suggests that TMEM16A may amplify ATP release from ISCs via connexin hemichannels due to changes either in cell volume or in membrane potential. This hypothesis does not oppose the conclusions from Wang et al., but rather puts a new perspective on an apparently complex bi-directional interplay between ATP release and TMEM16A activation. The discussion of the manuscript has now been revised accordingly.

2. The conclusion that this disruption of activity leads to reduction in refinement in the early auditory pathway is based entirely on physiological measurements and therefore no change in axonal projection is directly observed. It appears there might be other interpretations of their findings. I will focus primarily on the mapping of MNTB inputs to the LSO, which was achieved by glutamate uncaging – I think the configuration is making a slice that contains both the MNTB and the LSO (it would be great for authors to clarify this).

The following sentence was added to the Experimental Procedures section “in vitro electrophysiological recording and functional mapping”: From each mouse, one single brain slice was obtained which contained the MNTB and LSO to reach a maximum possible preservation of connections between these two nuclei.

Then a neuron in LSO is recorded in current clamp while glutamate uncaging occurs over the MNTB, driving individual neurons to fire action potentials. They find that MNTB neurons from a larger area all synapse onto a single LSO neuron, indicating a lack of refinement. That is one interpretation of these findings but there are many others, such as changes in local circuitry. The authors would need to do more experiments to reach this conclusion. For example, they could do experiments in voltage clamp to assay impact of excitatory vs inhibitory inputs to assay which is more responsible for membrane potential changes. Second they should use these method to demonstrate that they see a change in this distribution across development to make clear that this refinement occurs at the same ages that spontaneous activity is disrupted. Finally, the authors need to provide axon reconstructions to support the idea of the map refinement.

We agree with the reviewer that we do not provide anatomical proof of our conclusion. Unfortunately, this is currently beyond the scope of our possibilities, since Alena Maul, the author responsible for the slice recordings, left academia and her working group no longer exists. Therefore, we decided to tone down our conclusions.

Reviewer #3 (Recommendations for the authors):This is a well written, and well-presented paper. As described in the "Public Review" portion, most weaknesses are limited to interpretations of results and fitting the work more clearly into the existing literature, so as to not over-state the novelty of the work.Details:– In the abstract, the final sentence does not fit with existing literature and is not supported by the presented data. Previous papers from the Bergles group (Wang et al. 2015, Cell) indicate that purine release from cochlear inner supporting cells is upstream of TMEM16A. This line of the abstract, as well as lines 165-166, state the opposite. However, your data does not adequately disprove the Bergles lab hypothesis and does not address which part of the pathway initiates the calcium wave process. Omit these lines or provide stronger evidence.

In accordance to our reply to the Reviewer 3 public review, we followed this suggestion, and modified our statements:

“In addition, our study suggests a mechanism for the involvement of TMEM16A in the spreading of Ca^2+^ waves, which may also apply to other tissues expressing TMEM16A.” and “This supports the notion that TMEM16A is important for the spreading of spontaneous activity between ISCs of Kölliker’s organ, probably via P2 receptors.”

This will also include re-writing the title of the paper, as there is no evidence that TMEM16A itself initiates the wave patterns, although it is one molecule in the chain.

We followed this suggestion and changed the title accordingly:

“The Cl^-^-channel TMEM16A is involved in the generation of cochlear Ca^2+^ waves and promotes the refinement of auditory brainstem networks“.

– In figure 1 what was the total amount of time used for statistics? The 6 seconds of data shown is an extremely short time window, and not adequate to capture the variable timescales of the calcium waves.

The following information was added to the legend of Figure 1.

Figure 1G and H: “A time-lapse series of 1200 images with one image per second was analyzed.”

Figure 1I and J: “A time-lapse series of 400 images with one image per second was analyzed.”

– Line 113 – an extra space in the p-value.

Thank you, we corrected the typo (line133).

– Results section beginning line 154: Tritsch et al. 2007 reports the effect of suramin in reducing crenellations in inner supporting cells. In the same paper the crenellations were associated with calcium waves. Therefore, the experiment reported here is extremely similar to previously published work and this other work should be cited.

Thank you. The following reference was added:

“…confirming previous observations made by Tritsch et al. who found that Ca^2+^ signals are reduced upon application of suramin to cochlear explants from postnatal mice (Tritsch et al., 2007).”

– Figure 2 fonts are too small.

Thank you. The fonts were increased to improve readability.

– Please discuss how the changes in burst patterns detailed in figure 2 compare to previously published studies linking aberrant cochlear spontaneous activity with impaired MNTB-LSO tonotopic projections that use nearly identical techniques (Clause et al. 2014).– Line 207 – spell out approximately.

Thank you. We now spelled out “approximately”.

– MNTB neurons have higher thresholds, but ABR studies show normal thresholds. Please speculate as to why.

While we did not observe significant differences in threshold, there was a tendency toward increased ABR thresholds e.g. at 6 and 24 kHz and for clicks. The mean increase in hearing thresholds that we found in cKO mice using in vivo electrophysiology seems to contradict our ABR measurements that did not reveal significant differences in hearing thresholds between TMEM16A cKO mice and the wildtype littermates recorded at the same age. In general, there are two fundamental differences between these two techniques. First, in vivo electrophysiological recordings allow measuring the hearing thresholds of individual cells, whereas the ABR technique measures the sum response of all auditory neurons. Second, in vivo electrophysiological recordings show responses from a neuron within a defined nucleus of the auditory brain stem (in our case the MNTB). In contrast, the different peaks in the ABR waveform show responses generated by neurons from the level of the auditory nerve (wave I) to the superior olivary complex including MNTB, MSO and LSO (wave III). In P14 mice, waves IV and V are unclear and usually excluded from analysis due to the young age of the animals. Hence, we speculate that the differences in hearing thresholds that we detected in individual MNTB neurons using in vivo electrophysiology may be blunted by ABR measurements, which measure the summed response of many neurons from several auditory brain stem nuclei. Another possibility is that the cKO MNTB neurons recorded that were responding to tonotopically less constrained input showed increased thresholds for pure tones for this reason: i.e. were responding not just to the peak of the cochlear vibration but also to both tails. However, we cannot exclude other possible reasons for the discrepancy.

In response to the reviewer’s comment we now added a phrase to the discussion:

“We note that the discrepancy between significantly indistinguishable ABR thresholds and significantly elevated auditory thresholds of single MNTB neurons could arise for several reasons. […] However, other reasons cannot be excluded and a trend toward higher ABR thresholds was apparent for 6 and 24 kHz tone bursts.”

– Lines 283-284 would benefit from a very quick statistical test to demonstrate no difference between the CFs of MNTB neurons in the mutant and control groups.

Thank you. We tested with a two-tailed unpaired Student’s t-test, which demonstrates that there is no difference between CFs and MNTB neurons in the mutant and control group (p=0.51). This information is now also included in the manuscript.

– Re-write the sentence in lines 301-302 – perhaps move the word 'above' to after 'SPL'.

Thank you. The sentence was rephrased accordingly.

– Lines 306-307 – detail how many animals.

The information is now added at the beginning of the paragraph and reads:

“…the frequency response areas (FRAs) of single MNTB neurons were acquired in four cKO and four wildtype littermates…”

– Lines 307-309 appear to be totally speculative, if the statement remains it should be directly tested by comparing the firing rates of mutant and control MNTB neurons at the loudest sound levels tested.

The conclusion comes from the analysis of rate-level functions (two-way Anova test with Holm-Sidak multiple comparisons, effect of intensity p<0.001; effect of strain p<0.001; line 301-304, Figure 4G), and from the comparison of maximal firing rates (t-test, p=0.015; line 305-307, Figure 4H), which are often evoked by high intensity stimulation. However, the changes are notable already at 10 dB SPL and now we modified the statement accordingly:

“Apparently, MNTB neurons in cKO mice cannot achieve firing rates that are typically observed in wildtype littermates.”

– Lines 323-324 – briefly mention the illumination method here or in figure S6 legend.

The following was added: “For illumination a 405 nm continuous diode laser was used. Laser flashes were delivered through a light guide of 20 µm diameter which produced circular spots of 20 µm diameters in the focal plane. Laser pulses of 10 ms duration were delivered with 6 s delay time between uncaging sites.”

– Line 540 – I am guessing that you are actually using an upright, not an inverted microscope, if it has a dipping objective.

We are sorry for our mistake, which was corrected in the revised manuscript.

– Lines 554-556 are confusing and need to be re-worded.

The lines were updated as follows: “To be regarded as an event, volume changes had to match the following two criteria: First, events must show an increase DIC contrast greater than 10 % of the baseline level. […] Second, changes in fluorescence had to be larger than the surface area of one ISC.”

– Line 604 – please justify why the 2mV cutoff for a successful PSP was arbitrarily chosen. How does this relate to noise levels in the baseline?

Depolarizations from the recorded LSO neuron were only counted as PSPs if they coincided with the timepoint of glutamate uncaging. This was achieved by adding a second trace to the recording, visualizing the uncaging event in the MNTB (duration of the laser flash).

The 2 mV cutoff was chosen as LSO cells showed spontaneous firing of about 1-1.5 mV. This was done to avoid that spontaneous signals, which exactly coincide with the uncaging timepoint, were counted as signal.

**Author response image 1. sa2fig1:** (A-C) Example traces from the WT LSO cell for which the MNTB input map is shown in Figure S6D on the right. The upper part of each trace shows the response of the LSO cell to glutamate uncaging in the MNTB. The timepoints of glutamate uncaging are shown in the lower part. (A) 1.5 min recording time. (B) 26 s recording time. (C) 1.3 s recording time. The trace clearly shows a depolarization of the LSO cell following glutamate uncaging in the MNTB (left) as well as a spontaneous signal (right).

– Was there any change in the total size of the MNTB in TMEM16A KO animals?

We didn’t measure the total size of the MNTB (as a 3D structure). To place the virtual grid over the MNTB (i.e. for definition of the uncaging sites), we only visualized the MNTB in the focal plane. When we compare these cross-sectional areas between genotypes, the p-value is 0.05 with an unpaired t-test. For the analysis, the input widths and input areas were normalized to the cross-sectional area of the MNTB for each recording. Thus, differences in the MNTB cross-sectional area between genotypes should not affect the outcome of this experiment.

– Figure S6 – please add labeling to make the recording location (MNTB vs LSO neuron) more clear in the different panels.

In the Figure S6, we only show recordings of the MNTB neuron in A and B. D-E are schematic representations of the MNTBs with the glutamate uncaging sites that elicit responses in the LSO (LSO not shown). We are sorry that our figure legend was not clear. We revised it accordingly.

[Editors' note: further revisions were suggested prior to acceptance, as described below.]

The manuscript has been improved but there are some remaining issues that need to be addressed, as outlined below:For the most part, all reviewers felt that their concerns had been adequately addressed in the revision. There remains one issue regarding a point raised by Reviewers 1 and 3, and that has to do with the conclusions the authors reach regarding the mechanisms for the propagating calcium waves in ISCs. The authors conclude there is a "complex bi-directional interplay of TMEM16A and ATP receptors", but again the evidence the authors provide that TMEM16A influences ATP release is not apparent. Rather than there being "bidirectional signaling", the reviewers suggested that the authors should state that there is a positive feedback mechanism.

We now replaced “bidirectional signaling” by “positive feedback mechanism”, which nicely characterizes the relation between TMEM16A and purinergic receptors in ISCs.

The new text lines 382-404 is perhaps more confusing – the confusion coming from mixing in the result from other epithelial cells with what is known about ISCs in developing cochlea. A revision of this paragraph would help clarify models in the minds of non-experts.

We updated this section (now line 376-397) and now strictly separate findings for the cochlea and other epithelia in the discussion.

Reviewer #1 (Recommendations for the authors):The authors have addressed my concerns by weakening their interpretations.First, regarding the mechanisms for the propagating calcium waves in ISCs – the authors conclude "complex bi-directional interplay of TMEM16A and ATP receptors" but again the evidence it influences ATP release is limited – except through positive feedback mechanisms. Previous evidence for propagating waves in ISC (e.g. from the Bergles lab) should be stated more clearly.

Thank you for this suggestion. We agree and replaced “complex bi-directional interplay” by “positive feedback mechanisms” (e.g. line 385).

Furthermore, we clearly state that propagating Ca^2+^ waves were first reported by Tritsch et al. (lines 376-377).

Second, the only change made regarding the receptive field refinement is to call it "physiological refinement" with no discussion of potential mechanisms. Please clarify how you think frequency selectivity is altered.

Thank you. The reduced frequency-selectivity could result from superfluous functional projections from globular bushy cells (GBC) onto MNTB neurons (lines 409-414). We further briefly report on mechanisms identified in the visual system. Together with data from the auditory system this allows us to present a possible scenario, why tonotopical refinement might be compromised upon disruption of TMEM16A (lines 426-445).